# Near infrared sensitized exciton upconversion luminescence from inorganic perovskite nanocrystals

Yiyan Zhang[1,2], Tianyu Zhao[1,2], Yuming Deng[3], Xinyue Liu[3], Xiaorong Zhang[1,2], Tong Zhu ●[3] ✉, Hans Ågren[1,2,4] & Guanying Chen ●[1,2,5] ✉

Lead halide perovskites serve as an important class of photoelectrical materials in modern technological applications, such as light emitting diodes, photodetectors, and solar cells. However, the inability to respond to near infrared light poses a tight constraint on their performances. This study reports a class of broadband near infrared-responsive inorganic lead halide perovskite nanocrystals, which emit a palette of tunable upconversion luminescence via organic dye-lanthanide tandem sensitization. The coordination of dyes (IR783) to the surface of ytterbium-doped cesium lead halide nanocrystals entails an intense and broad spectral response range of near infrared light (600-860 nm). Sub-10 Wcm$^{-2}$ light irradiance at 804 nm induced ultrabright exciton luminescence (tunable from 520 to 625 nm), about 27,500 folds brighter than the one without dye sensitization, with upconversion brightness reaching 3.22 M$^{-1}$cm$^{-1}$. Transient absorption spectra revealed an ultrafast triplet energy transfer process ($9.28 \times 10^8$ s$^{-1}$) from dyes to ytterbium dopants with near-unity efficiency (98.4%), followed by cooperative sensitization that excites delocalized excitons. These broadband near infrared-responsive inorganic perovskite nanocrystals have implications for applications ranging from solar cells to near infrared imaging and sensing.

Lead halide perovskites have garnered significant attention in both fundamental research and technological applications due to their large molar extinction coefficient, high carrier mobility, and material abundance. In particular, they found prevalent uses in perovskite solar cells with efficiencies comparable to market-dominating silicon solar cells but at much lower production cost[1]. Nanoscale lead halide perovskites, such as cesium lead halides (CsPbX$_3$, X=Cl, Br, and I) inorganic perovskites nanocrystals with a minimum bandgap of 1.73 eV (when X = I)[2], inherit the advantages of their bulk counterparts while offering unique properties like excellent thermal stability, high photoluminescence efficiency, and narrow tunable emissions.

These intriguing features have spurred a broad spectrum of photonics and biophotonics applications, such as light-emitting diodes, photodetectors, photocatalalyzers, nanoscintillators, immunolabelling, and solar cells[2-8], where inorganic perovskite nanocrystals play a central role in harnessing incident photons and generating high-efficiency luminescence. Despite these promises, the inability of lead halide perovskites to respond to near infrared (NIR) light—which constitutes over 40% of terrestrial sun irradiance spectrum—pose a significant limit on optoelectronic device performances and prevent their uses in NIR-responsive biophotonics, photocatalysis, and water splitting[1,9-12].

[1]MIIT Key Laboratory of Critical Materials Technology for New Energy Conversion and Storage, School of Chemistry and Chemical Engineering, Ministry of Education, Harbin Institute of Technology, 150001 Harbin, People's Republic of China. [2]Key Laboratory of Micro-systems and Micro-structures, Ministry of Education, Harbin Institute of Technology, 150001 Harbin, People's Republic of China. [3]Laser Micro/Nano Fabrication Laboratory, School of Mechanical Engineering, Beijing Institute of Technology, Beijing 100081, P. R. China. [4]Department of Physics and Astronomy, Division of X-ray Photon Science, Uppsala University SE-, 75121 Uppsala, Sweden. [5]Harbin KY Semiconductor, Inc., Harbin, China. ✉e-mail: tongzhubit@bit.edu.cn; chenguanying@hit.edu.cn

Photon upconversion, the process that enables spectral conversion of low energy light into high energy photons, offers a promising approach to addressing this limitation. Multiphoton NIR absorption in lead perovskites has been shown to generate visible upconversion luminescence and lasing[13–16]. Yet, this nonlinear process requires ultrafast intense pulsed laser to produce ultrahigh laser power density ( >$10^6$ Wcm$^{-2}$), limiting practical applications. Alternatively, mixing halide perovskite nanocrystals or constructing heterostructures with lanthanide upconverting fluoride nanocrystals can produce upconversion luminescence under low NIR laser power density ~980 nm, 10 Wcm$^{-2}$[17–19]. However, the narrow (10 nm) and low absorption cross section ($10^{-20}$ cm$^2$) of NIR-harvesting lanthanide dopants and the insulation nature of involved fluoride lattice preclude their uses in photoelectrical applications. Triplets from halide perovskite nanocrystals are able to sensitize diphenol anthracene to produce blue triplet-triplet annihilation upconversion luminescence under photoexcitation at 532 nm (100 W cm$^{-2}$)[20], but the limitation lies in that the absorption stems from halide perovskite nanocrystals which is still in the visible region. Achieving photon upconversion in lead halide perovskite nanocrystals with responsiveness to broadband NIR light remains an unresolved challenge.

It has been implied that there exist strong lanthanide-exciton interactions in lanthanide-doped lead halide perovskite nanocrystals such as ytterbium (Yb$^{3+}$)-doped CsPbCl$_3$ nanocrystals, in which a single ultraviolet-excited exciton generates two photons at 980 nm (from Yb$^{3+}$ dopants) with a quantum cutting efficiency >170%[21,22]. In addition, small organic molecules attached to the surface of sodium yttrium fluoride (NaYF$_4$) lanthanide nanoparticles are able to sensitize a set of lanthanide dopants (Yb$^{3+}$, Pr$^{3+}$, Nd$^{3+}$, etc.) and produce broadband-sensitized lanthanide luminesence[23–27].

Here, we show that coordinating small NIR organic dyes to the surface of lanthanide-doped lead halide perovskite nanocrystals could enable tandem sensitization of excitons, leading to intense excitonic upconversion luminescence under low irradiance and with broadband response to NIR light (Fig. 1). The surface-coordinated dyes (IR783) of ytterbium-doped cesium lead halide nanocrystals entails an intense and broad absorption ranging from 600 to 860 nm. Sub-10 Wcm$^{-2}$ light irradiance at 804 nm induced ultrabright exciton upconversion luminescence (tunable from 520 to 625 nm), about 27,500 folds brighter than the one without dye sensitization, with upconversion brightness reaching 3.22 M$^{-1}$cm$^{-1}$.

## Results

### Design of dye-sensitized upconverting perovskite nanocrystals

To demonstrate this idea, we chose to construct dye-sensitized upconverting perovskite nanocrystal (UCPN) system using inorganic CsPbX$_3$ perovskite nanocrystals (doped with Yb$^{3+}$ ions) capped with oleic acid ligands, and small organic cyanine dye molecules (IR783) containing functionalized sulfonic groups as light-harvesting antennas (Fig. 1). Yb$^{3+}$ ion is selected as model lanthanide ion here, as it has been shown to have a strong interaction with delocalized excitons in lead halide perovskites[22,28,29], and can be used as an effective energy acceptor due to large absorption cross section and broad absorption range compared to other lanthanides. The Yb$^{3+}$ ions could be incorporated into the nanocrystal lattice by substituting the divalent lead (Pb$^{2+}$) cation at the B site in the ABX$_3$ (CsPbX$_3$) perovskite structure, forming two types of octahedra with Yb$^{3+}$ and Pb$^{2+}$ as central ions. The functionalized sulfate groups of IR783 dye molecules enable stable coordination to the cationic sites on the nanocrystal surface by partially replacing the oleic acid ligands of the as-synthesized CsPbX$_3$ nanocrystals. The surface-anchored IR783 antenna molecules efficiently harvest NIR incident light over a broad range and interact with neighboring [YbX$_6$]$^{3-}$ octahedra, facilitating energy transfer from IR783 molecules to surface Yb$^{3+}$ ions. Ultimately, two excited Yb$^{3+}$ ions cooperatively transfer energy to one surrounding valence electron to excite a delocalized exciton, thereby producing band-edge exciton upconversion luminescence.

### Characterization of the dye-sensitized upconverting perovskite nanocrystal system

Undoped and doped CsPbBr$_3$ perovskite nanocrystals were synthesized through a typical hot-injection method with small modifications[21,22,30,31]. Representative transmission electron microscopy (TEM) image showed that the resulting CsPbBr$_3$: Yb$^{3+}$ perovskite nanocrystals are monodisperse with an average size of 11.45 ± 0.05 nm (Fig. 2a, and Supplementary Fig. 1). Furthermore, high angle annular dark field scanning electron microscopy image and energy dispersive x-ray spectroscopy analysis proved that the doped Yb$^{3+}$ ions are homogeneously distributed in the CsPbBr$_3$ nanocrystals (Supplementary Fig. 2). X-ray diffraction patterns of the CsPbBr$_3$:Yb$^{3+}$ nanocrystals indicated that doping of Yb$^{3+}$ ions does not alter the crystal phase of the perovskite nanocrystals (Supplementary Fig. 3).

Compared to the absorption spectrum of untreated perovskite nanocrystals, attachment of small IR783 dye to the CsPbBr$_3$ nanocrystal

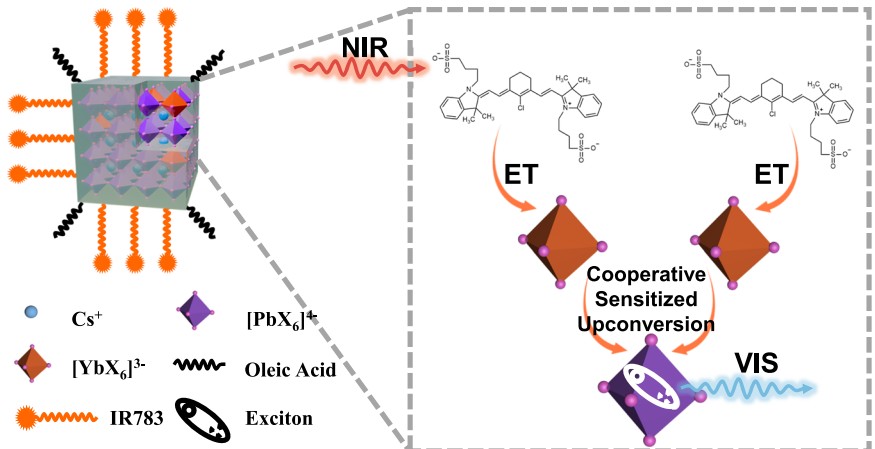

**Fig. 1 | A diagram illustrating the dye-perovskite nanocrystal coupled system with cooperative sensitized upconversion.** The cyan sphere represents the A-site ion (Cs$^+$), while the purple and dark brown octahedra correspond to [PbX$_6$]$^{2-}$ and [YbX$_6$]$^{3-}$, respectively, in the ABX$_3$ (CsPbX$_3$) perovskite structure (semi-transparent light green cube). The black wavy lines depict the oleic acid ligands, while the yellow wavy lines (with star ends) represent the IR783 dye molecules (chemical structure shown on the right) attached to the nanocrystal surface. (ET: energy transfer, NIR: near infrared, VIS: visible).

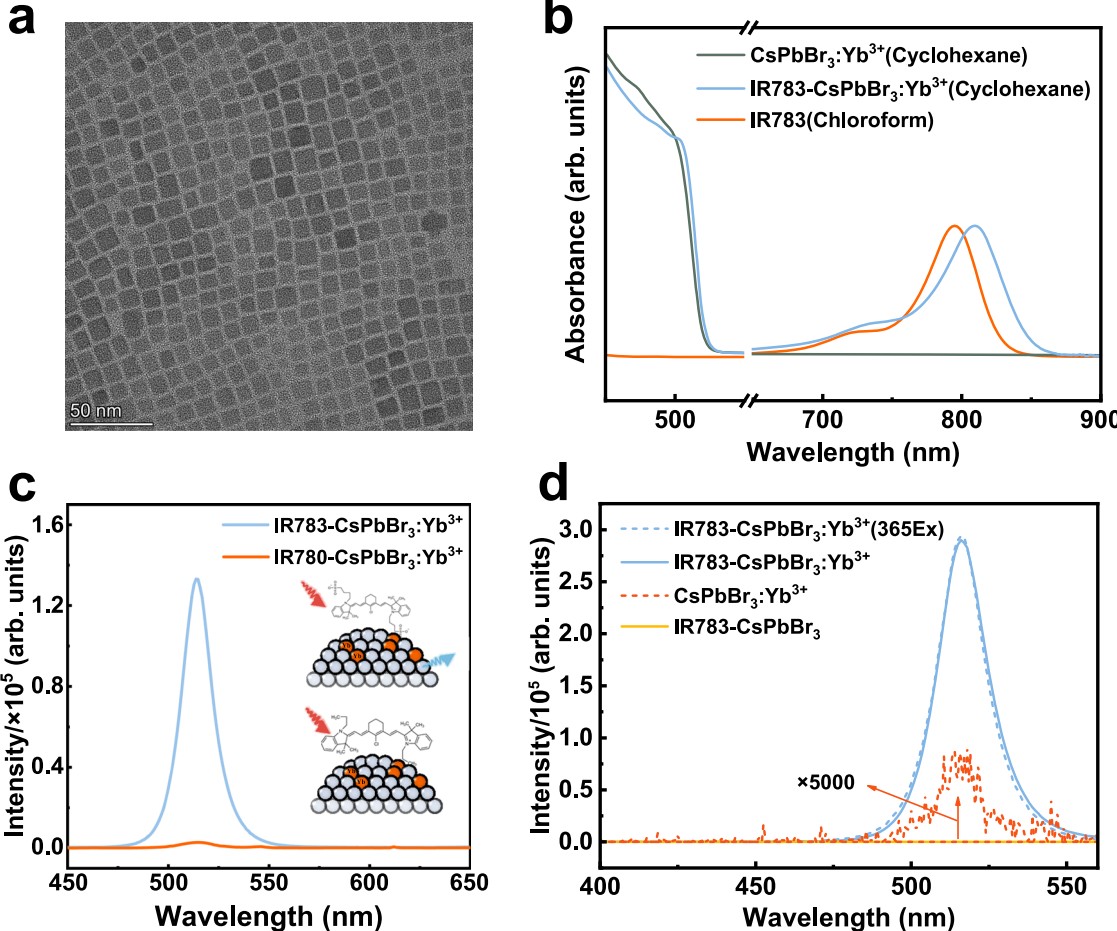

**Fig. 2 | Characterizations of the dye-sensitized upconverting perovskite nanocrystal system. a** Representative TEM image of CsPbBr$_3$:Yb$^{3+}$ nanocrystals. **b** Absorption spectra of CsPbBr$_3$:Yb$^{3+}$ nanocrystals (in hexane), IR783-CsPbBr$_3$:Yb$^{3+}$ nanocrystals (in hexane) and pure IR783 dye (in chloroform). **c** Upconversion luminescence spectra of IR783-CsPbBr$_3$:Yb$^{3+}$ and IR780-CsPbBr$_3$:Yb$^{3+}$ nanocrystals; the inset depicts a schematic of IR783 and IR780 molecules on the surface of CsPbBr$_3$:Yb$^{3+}$ nanocrystals. **d** Upconversion luminescence spectra of CsPbBr$_3$:Yb$^{3+}$ nanocrystals (red solid line), IR783-CsPbBr$_3$ nanocrystals (orange solid line), IR783-CsPbBr$_3$:Yb$^{3+}$ nanocrystals (blue solid line), in contrast to the Stokes luminescence spectrum of IR783-CsPbBr$_3$:Yb$^{3+}$ nanocrystals under 365 nm excitation (blue dashed line). Dye-sensitized upconversion spectra from IR783-CsPbBr$_3$:Yb$^{3+}$ and IR780-CsPbBr$_3$:Yb$^{3+}$ nanocrystals were acquired under 804 nm continuous-wave laser irradiance at 8.4 W/cm$^2$, while upconversion spectra from CsPbBr$_3$:Yb$^{3+}$ nanocrystals were acquired under 980 nm continuous-wave laser irradiance at 8.4 Wcm$^{-2}$.

surface entails appearance of an intense wide absorption band ranging from 600 nm to 860 nm, nearly identical to the one of pure IR783 dye molecules in chloroform but with a red shift of about 10 nm (Fig. 2b)[32] This red-shift arises from the fact that the IR783 dye (chloroform) and the IR783-CsPbBr$_3$:Yb$^{3+}$ dye-perovskite coupled system (cyclohexane) were measured in different solvents[33]. Since pure IR783 dye molecule shows no absorption when dissolved in cyclohexane, observation of its absorption in cyclohexane indicated its successful attachment to the CsPbBr$_3$ nanocrystal surface, probably through chemical coordination of its two functionalized sulfonic groups to the surface cations of nanocrystals (Supplementary Fig. 4). The result from FTIR spectroscopy further proved the attachment (Supplementary Fig. 5). To confirm this, we utilized IR780 to substitute IR783 molecules, with identical molecular structure but without the two sulfonic groups, to sensitize CsPbBr$_3$:Yb$^{3+}$ nanocrystals (Fig. 2c, d)[34]. The absorption spectrum of IR780-CsPbBr$_3$:Yb$^{3+}$ (Supplementary Fig. 6) indicated that IR780 has been adsorbed to the nanocrystals. Negligible upconversion luminescence was observed, verifying the importance of chemical interaction between the perovskite nanocrystals and the surface IR783 molecules.

IR783 dye-sensitized CsPbBr$_3$:Yb$^{3+}$ nanocrystals exhibit an intense upconversion luminescence peaked at 515 nm under continuous-wave laser irradiance at 804 nm, which exactly overlaps with that of band-edge exciton luminescence from CsPbBr$_3$:Yb$^{3+}$ nanocrystals under 365 nm excitation (Fig. 2d). This fact verifies that the sensitized upconversion luminescence stems from delocalized excitons. In sharp contrast, no upconversion luminescence was observed in the IR783-CsPbBr$_3$ control group (excited by 804 nm laser, power density= 8.4 Wcm$^{-2}$), confirming the importance of Yb$^{3+}$ dopants to induce exciton upconversion luminescence. We varied the Yb$^{3+}$ doping concentration and found that the upconversion luminescence is highly dependent on the doping concentration, with an optimized nominal concentration of about 40% while the actual doping concentration is 0.8% (Supplementary Fig. 7, Supplementary Table 1). Note that doping Yb$^{3+}$ ions into CsPbBr$_3$ nanocrystals can induce weak upconversion luminescence under continuous-wave laser irradiance at 980 nm, corresponding to the $^2F_{7/2} \rightarrow {}^2F_{5/2}$ transition of Yb$^{3+}$ ions. This indicates that lanthanide-exciton interaction is critical in realizing exciton upconversion luminescence from dye-sensitized UCPNs. Compared to that of pure the CsPbBr$_3$: Yb$^{3+}$ nanocrystals, the upconversion intensity of dye-sensitized UCPNs increased by more than 27,500 folds with an upconversrion luminescence quantum yield of $1.23 \times 10^{-3}$ % (excited at 804 nm, power density= 8.4 Wcm$^{-2}$). The procedure for determining enhancement factor was described in Supplementary Figs. 8 and 9. Though this luminescence quantum yield is small, the strong absorption of IR783 dye impart a decent upconversion

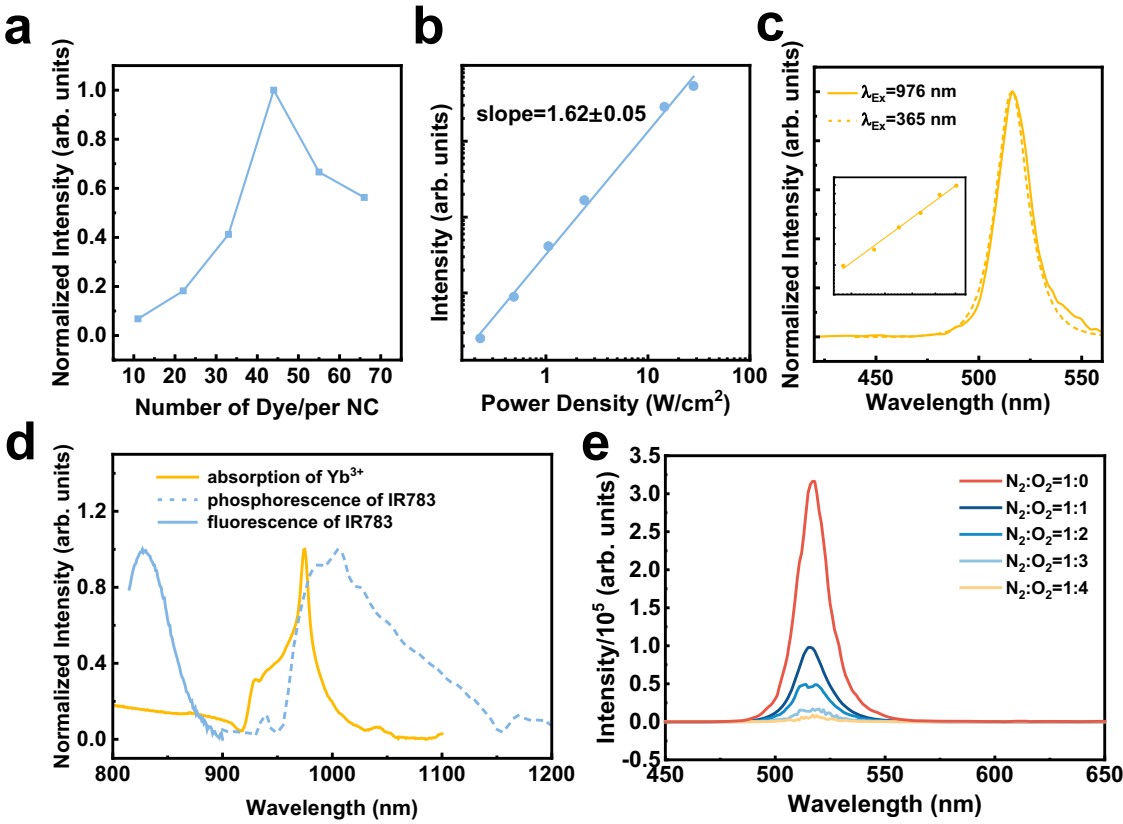

**Fig. 3 | Mechanistic investigations on upconversion dynamics in dye-sensitized perovskite nanocrystals. a** IR783 density dependent upconversion luminescence spectra of IR783-CsPbBr$_3$:Yb$^{3+}$ nanocrystals. **b** Power density dependent upconversion luminescence intensity of IR783-CsPbBr$_3$:Yb$^{3+}$ nanocrystals; excited at 804 nm. **c** Upconversion (solid line) and Stokes (dashed line) luminescence spectra of CsPbBr$_3$: Yb$^{3+}$ nanocrystals; inset shows the corresponding relation between power density and upconversion luminescence intensity; excited at 980 nm. **d** Absorption spectra of Yb$^{3+}$ ions (orange solid line), contrasted with fluorescence (blue solid line) and phosphorescence (blue dashed line) spectra of IR783 dye. **e** Upconversion luminescence spectra of IR783-CsPbBr$_3$:Yb$^{3+}$ in atmospheres with varied N$_2$/O$_2$ ratios; excited at 804 nm, power density= 8.4 Wcm$^{-2}$. All nanocrystals were dispersed in hexane in a cuvette during spectroscopic characterizations.

brightness (B) to IR783 dye-sensitized CsPbBr$_3$:Yb$^{3+}$ nanocrystal system, which can be estimated following the formula (1)[35]:

$$B = \varepsilon \times QYs \tag{1}$$

where B is defined as the product of the molar absorption coefficient ($\varepsilon$) and the luminescence quantum yield of the sample. We calculated the upconversion brightness reach as high as 3.22 M$^{-1}$cm$^{-1}$ under sub-10 Wcm$^{-2}$, comparable that of a few well-established one-photon NIR fluorescent dyes and superior to that of common upconverting materials (Supplementary Table 2).

### Mechanistic investigations on upconverting dynamics in dye-sensitized perovskite nanocrystals

To achieve a better understanding of the results, we optimized the density of IR783 molecules on the CsPbBr$_3$:Yb$^{3+}$ nanocrystal surface to enhance upconversion luminescence. As shown in Fig. 3a, the optimal density of IR783 was determined to be about 41 molecules per nanocrystal, as detailed in Methods and Supplementary Fig. 10. Beyond this optimal dye loading, aggregation induced quenching between cyanine dye molecules was observed[36,37], leading to decreased upconversion luminescence. A power density dependent upconversion luminescence measurement revealed a slope of 1.62 ± 0.05, indicating a two-photon process for dye-sensitized exciton upconversion luminescence (Fig. 3b, Supplementary Fig. 11). Furthermore, direct laser excitation of Yb$^{3+}$ ions at 976 nm induced two-photon upconversion luminescence from CsPbBr$_3$: Yb$^{3+}$ nanocrystals, with spectra identical to band-edge Stokes exciton luminescence (Fig. 3c, and Supplementary Fig. 12) and

dye-sensitized upconversion luminescence (Fig. 2d). It is noteworthy that the discrepancy of the spectra in Fig. 3c at ~550 nm may be caused by the potential Er$^{3+}$ or Tb$^{3+}$ contamination. Because Yb$^{3+}$ dopants have a single excited energy level of the $^2$F$_{5/2}$ state ( ~980 nm), the two-photon nature of the upconverted exciton luminescence at 515 nm and the energy conservation principle necessitate the cooperative energy transfer from two excited Yb$^{3+}$ dopants to excite a single exciton.

It is noteworthy that the fluorescence and phosphorescence of IR783, measured at both room temperature and 77 K, partially overlap with the absorption of Yb$^{3+}$ ion (specifically the $^2$F$_{7/2}$ → $^2$F$_{5/2}$ transition), facilitating effective nonradiative energy transfer (Fig. 3d). The energy transfer from IR783 cyanine dye to Yb$^{3+}$ can occur through either the singlet or triplet excited states[25]. Molecular dioxygen (O$_2$) is known to efficiently quench triplet states but produce no effects on the singlet states[36,38]. To determine singlet or triplet energy transfer from IR783 dye to Yb$^{3+}$ here, the environment was controlled by adjusting the ratio of N$_2$ and O$_2$ injected into a quartz cuvette (Fig. 3e, excited at 804 nm, power density= 8.4 Wcm$^{-2}$). The results demonstrated that as the O$_2$ concentration increased, the upconversion luminescence of IR783-CsPbBr$_3$:Yb$^{3+}$ nanocrystals rapidly decreased. Conversely, O$_2$ molecules have negligible influence on the Stokes luminescence of excitons in CsPbBr$_3$ nanocrystals (Supplementary Fig. 13). Additionally, when IR783-CsPbBr$_3$:Yb$^{3+}$ nanocrystals prepared in a glovebox were exposed to air, the upconversion luminescence diminished significantly as exposure time increased (Supplementary Fig. 14). Taken together, these findings support the importance of triplets in sensitizing Yb$^{3+}$ dopants to produce cooperative upconverted exciton luminescence.

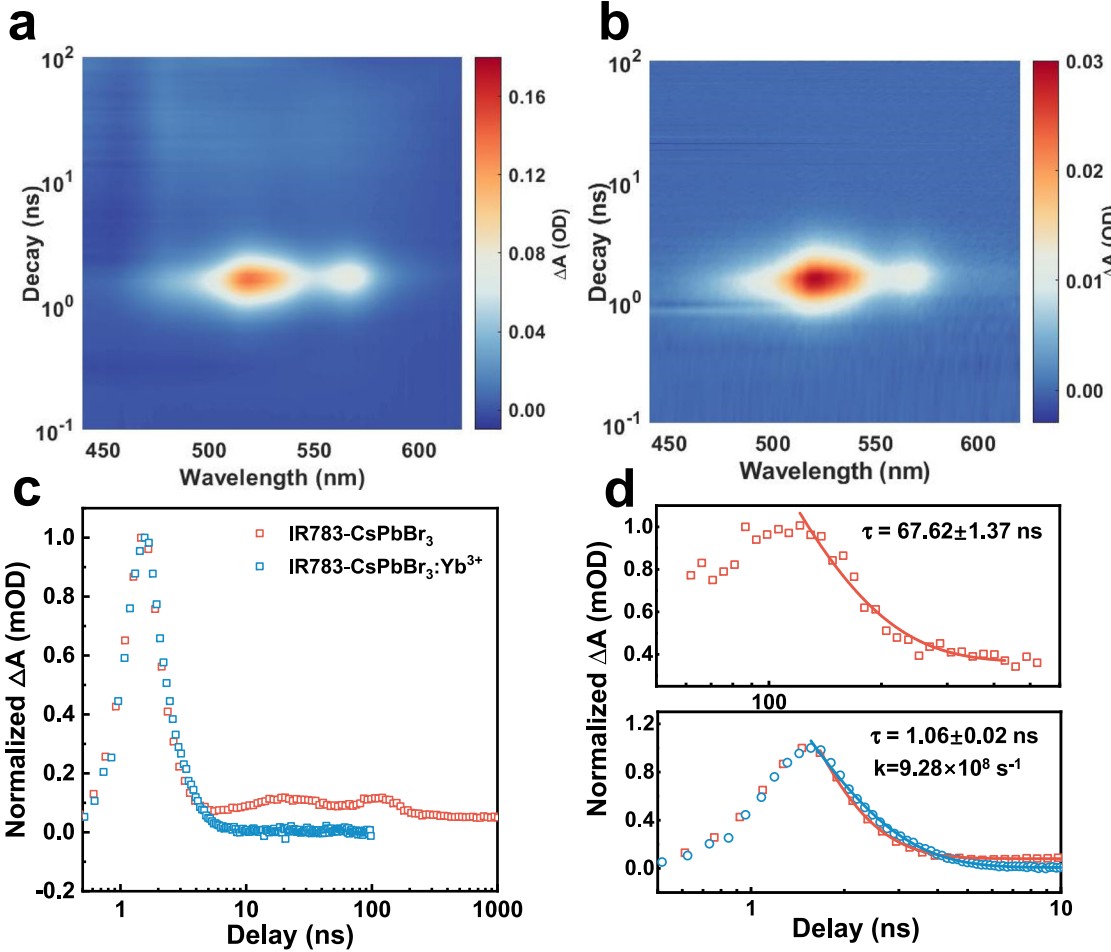

**Fig. 4 | Triplet energy transfer dynamics from IR783 to CsPbBr₃:Yb³⁺ perovskite nanocrystals. a** Transient absorption spectrum of IR783-CsPbBr₃ nanocrystals. **b** Transient absorption spectrum of IR783-CsPbBr₃: Yb³⁺ nanocrystals. **c** TA decay of IR783-CsPbBr₃ nanocrystals and IR783-CsPbBr₃: Yb³⁺ nanocrystals over the entire measured timescale. **d** Triplet decay (top) and singlet decay (bottom) of IR783-CsPbBr₃ nanocrystals compared with the decay of IR783-CsPbBr₃: Yb³⁺ nanocrystals; the solid curves represent the corresponding single-exponential fitting results of the decay processes, detected at 522 nm, power density =1.58 μJ cm⁻².

To investigate the triplet energy transfer (TET) kinetics between IR783 and Yb³⁺ dopants in CsPbBr₃ nanocrystals, nanosecond transient absorption (ns-TA) measurements were conducted under 750 nm ns-laser excitation (Fig. 4). To rule out the orientation and distribution effects of IR783 dye on nanocrystals, IR783-CsPbBr₃ was utilized as control sample. Limited by the large overlap between the singlet and triplet signals over the wavelength, the singlet and the triplet spectra of IR783 were differentiated over timescale. As shown in the spectra, singlet state (Fig. S15a) presented strong wide excited-stated absorption peaked at 522 nm and 560 nm while triplet state (Fig. S15b) absorbed intensely at 480 nm and 522 nm, in accordance with the results from reported paper[32]. In the case of IR783-CsPbBr₃: Yb³⁺, however, the long-lived component is absent (Supplementary Fig. 16). Transient absorption spectrum of IR783-CsPbBr₃ control sample shows an intense excited state absorption (ESA) peaked at 522 nm (Fig. 4a), which consists of both the singlet and triplet component of IR783 that can be time-resolved at two distinct stages (Fig. 4c)[39]. At first 5 ns, singlet component dominates the ESA process, in which a fast luminescence decay with a lifetime of 0.75 ± 0.02 ns was acquired (Supplementary Fig. 17), consistent with the ground state bleaching lifetime (0.82 ± 0.12 ns) at 800 nm (Supplementary Fig. 18), and singlet fluorescence decay lifetime (0.73 ± 0.01 ns) (Supplementary Fig. 19). After depletion of the singlet state, a long-lived component emerged in the second stage, corresponding to the T₁-to-Tₙ excited state absorption of IR783. The triplet lifetime was estimated to be ~67.62 ± 1.37 ns

(Fig. 4d). This shortened triplet lifetime, compared to other cyanine dyes[36,40], may be attributed to the pronounced heavy-atom effect induced by the surrounding abundance of Cs⁺ and Pb²⁺ ions. Figure 4b shows the transient absorption spectrum of IR783-CsPbBr₃:Yb³⁺ following 750 nm laser excitation, revealing a fast decay of ESA process at 522 nm. In the presence of Yb³⁺, the estimated lifetime is about ~1.06 ns, slightly longer than the singlet lifetime of 0.75 ns in IR783-CsPbBr₃, which indicates the observed exclusive TA signal is probably a mixture of both singlet and triplet signal with close lifetimes. This measured lifetime was used to estimate the triplet lifetime. The triplet state lifetime was estimated to be 1.06 ± 0.02 ns, corresponding to a TET rate from IR783 to Yb³⁺ of 9.28 × 10⁸ s⁻¹, with a near-unity TET efficiency of 98.4% (Fig. 4d).

It has been reported that Yb_Pb defects tends to appear in pairs owing to charge compensation, causing (2Yb_Pb+V_Pb)⁰ defect complexes[41]. Besides, the angle configuration is more favorable than the linear configuration (Fig. 5a) according to DFT calculations[42]. The observation of cooperative upconversion luminescence from Yb³⁺-dimers in perovskite nanocrystals (CsPbCl₃:Yb³⁺) gives evidence for the existence of such defects (Supplementary Fig. 20). This structural feature may enhance the cooperative sensitized upconversion in dye-sensitized Yb³⁺-doped perovskite nanocrystals. A clear legible depiction of the exciton upconversion process is shown in Fig. 5b. In this process, IR783 molecules act as broadband NIR light absorbers on the surface of CsPbBr₃:Yb³⁺ nanocrystals, followed by highly accelerated

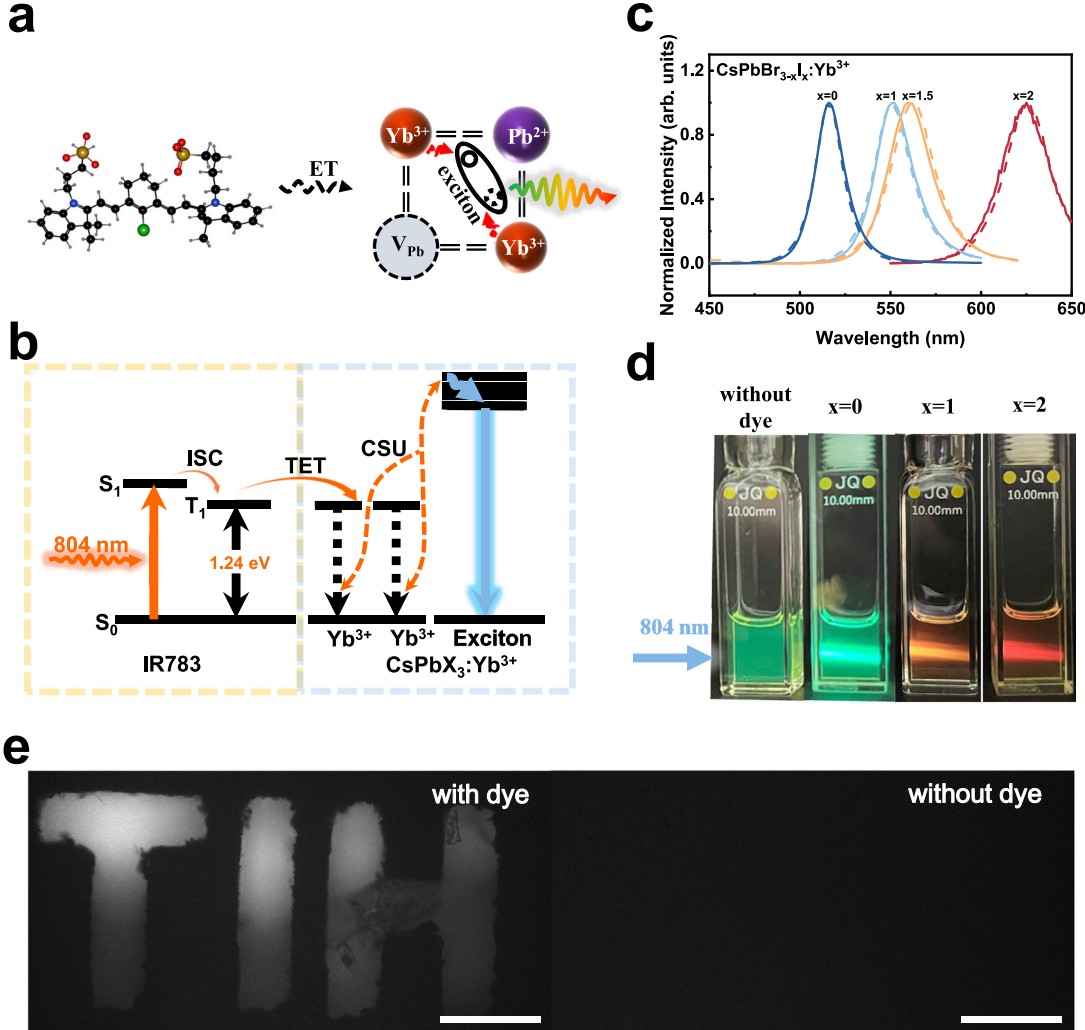

**Fig. 5 | Multicolor exciton upconversion luminescence from dye-sensitized perovskite nanocrystals. a** A schematic illustration of triplet energy transfer from dye to paired $Yb^{3+}$ dopants in preferred configuration to elicit exciton upconversion. **b** Simplified energy level diagram illustrating the IR783 molecule, $Yb^{3+}$ ions, and $CsPbBr_3$ semiconductor bands, along with the upconverting pathways. The diagram includes the involved energy transfer processes (solid wavy yellow arrow), the cooperative sensitization process (dashed wavy yellow arrow), and non-radiative multiphonon relaxations (blue wavy arrow), all contributing to the exciton upconversion luminescence (blue solid arrow). **c, d** Multicolor exciton upconversion luminescence spectra (solid line) (**c**) and corresponding upconverting photographic images (**d**) from IR783-senstized $CsPbBr_{3-x}I_x$: $Yb^{3+}$ ($x = 0, 1, 1.5,$ and 2) and non-sensitized $CsPbBr_3$:$Yb^{3+}$ nanocrystals; the corresponding Stokes luminescence spectra from excitons are also included for reference (under 365 nm excitation). **e** Microscopic upconversion luminescence images of $CsPbBr_3$: $Yb^{3+}$ patterns with (left) and without (right) IR783 sensitization; scale bar, 100 μm. Excited at 804 nm; power density of 8.4 $Wcm^{-2}$ for c and d, and of 416.7 $Wcm^{-2}$ for (**e**). (ET: energy transfer, $S_0$: ground state, $S_1$: excited singlet state, $T_1$: excited triplet state, ISC: intersystem crossing, TET: triplet energy transfer, CSU: cooperative sensitized upconversion).

intersystem crossing to triplet states, which implemented near-unity TET to the doped $Yb^{3+}$ ions. The configuration of the $(2Yb_{Pb}+V_{Pb})^0$ defect in the $CsPbBr_3$: $Yb^{3+}$ nanocrystals facilitate the achievement of cooperative sensitized exciton upconversion.

## Multicolor exciton upconversion luminescence from dye-sensitized perovskite nanocrystals

Inspired by the broadband sensitized exciton upconversion luminescence from dye-sensitized $CsPbBr_3$:$Yb^{3+}$ nanocrystals, we hypothesized that bandgap engineering of lead halide semiconductor nanocrystals could enable tunable multicolor exciton upconversion via the same tandem sensitization mechanism.

To test this hypothesis, we synthesized a series of $CsPbBr_{3-x}I_x$: $Yb^{3+}$ ($x = 0, 1, 1.5,$ and 2) perovskite nanocrystals with mixed halides to achieve tunable semiconductor bandgaps, and chemically coordinated the IR783 dye to the nanocrystal surface (Fig. 5c). As anticipated, tunable upconversion luminescence

spectra were realized, with visualized upconversion color shifting from green to orange and then to red under 804 nm excitation (Fig. 5d, power density= 8.4 $Wcm^{-2}$). The upconversion luminescence spectra matched perfectly with the Stokes luminescence spectra of the perovskite nanocrystals excited under ultraviolet light (365 nm), confirming that the observed upconverted luminescence also originates from band-edge excitons. Power dependence studies of upconversion spectra confirmed a two-photon process involved to produce multicolor upconversion, aligning with the demonstrated dye-lanthanide tandem sensitization mechanism (Supplementary Fig. 21). These broadband NIR-responsive lead halide nanocrystals are promising for upconversion luminescence imaging. A pattern "TIH", made by IR783-$CsPbBr_3$:$Yb^{3+}$ nanocrystals, glowed clearly under 804 nm light irradiance, in sharp contrast with the same pattern made from $CsPbBr_3$:$Yb^{3+}$ nanocrystals (Fig. 5e, excited at 808 nm, power density= 416.7 $Wcm^{-2}$).

## Discussion

In this work, we have extended the responsive wavelength of inorganic lead halide perovskite nanocrystals into the near-infrared region by constructing a dye-perovskite nanocrystal coupled system. Leveraging the near-unity triplet energy transfer between $Yb^{3+}$ dopants incorporated at lead halide perovskite lattice and the chemically coordinated IR783 antenna molecules at the nanocrystal surface, the system exhibited intense cooperative sensitized upconversion luminescence of band-edge excitons, with absorption across a broad range from 600 to 860 nm. Dye sensitization enhanced the exciton upconversion luminescence intensity by more than 27,500-fold under sub-10 $Wcm^{-2}$ continuous-wave laser irradiance at 804 nm, reaching upconversion brightness of 3.22 $M^{-1}cm^{-1}$. Additionally, precise multicolor exciton upconverted luminescence could be realized by bandgap engineering of lead halide perovskite nanocrystals through halide mixing. The stability of this system under different conditions was further evaluated (Supplementary Figs. 22–24). Although the stability of the materials needs further improvement, the dye sensitizing strategy is ideal to accomplish bright upconversion luminescence of PeNCs. Also, the quantum yield may be further enhanced by modifying the dye molecule and optimizing the distance between dye and $Yb^{3+}$. This work reveals a strategy to extend the absorption wavelength of PeNCs. The findings have significant implications for enhancing the efficiency of perovskite photovoltaic devices and expanding their applications in fields such as near-infrared imaging, photodetection, and photocatalysis.

## Methods

### Materials

Lead acetate trihydrate ($Pb(OAc)_2 \cdot 3H_2O$) (99.998%, Aladdin), ytterbium acetate tetrahydrate ($Yb(OAc)_3 \cdot 4H_2O$) (99.9%, Aladdin), chlorotrimethylsilane (TMS-Cl) (98%, Aladdin), Bromotrimethylsilane (TMS-Br) (98%, Aladdin), Iodotrimethylsilane (TMS-I) (97%, Aladdin), 1-octadecene (ODE) (90%, Sigma Aldrich), oleylamine (OAm) (70%, Sigma Aldrich), oleic acid (OA) (90%, Sigma Aldrich), anhydrous ethanol (99.5%, Aladdin), isopropyl alcohol (99.5%) and cyclohexane (99.5%, Aladdin) were used without further treatment.

### Synthesis of $Yb^{3+}$-doped $CsPbBr_3$ inorganic perovskite nanocrystals

Typically, CsOAc (0.05 mmol), $Pb(OAc)_2 \cdot 3H_2O$ (0.2 mmol), $Yb(OAc)_3$ (0.1 mmol), oleic acid (1 mL), OAm (0.5 mL), ODE (5 mL) to a 50 mL three-neck flask. The mixture was stirred and purged with nitrogen at room temperature for 5 min, then heated to 120 °C and degassed for 60 min. When the temperature reached 190 °C, 0.2 mL of TMS-Br was rapidly injected into the flask. The reaction was immediately quenched by cooling the flask with ice water. The resulting crude solution was centrifuged at 8000 rpm for 5 min, the supernatant was discarded, and the precipitate was redispersed in cyclohexane. The solution was then centrifuged again at 6000 rpm for 5 min, and the supernatant was collected.

### Synthesis of $Yb^{3+}$-doped $CsPbBr_{3-x}I_x$ inorganic perovskite nanocrystals

The synthesis followed the abovementioned protocol except the TMS-Br was replaced with mixed solution with different halide precursor ratio of TMS-Br/TMS-I.

### Synthesis of dye-sensitized upconversion perovskite nanocrystals

The IR783-perovskite nanocrystal complexes were prepared by adding IR783 powder to a solution of perovskite nanocrystals in cyclohexane, followed by sonication for a controlled duration (5–30 min). The mixture was then filtered to obtain a clear solution containing the IR783-perovskite nanocrystal complexes. Given the negligible solubility of IR783 in cyclohexane, it was assumed that all IR783 molecules were anchored to the surfaces of the perovskite nanocrystals dispersed in hexane.

### Characterizations

The size and morphology of the resulting perovskite nanocrystals were characterized by transmission electron microscopy (TEM, Tecnai G2 Spirit Twin 12). High-resolution TEM (HRTEM), high angle annular dark field scanning transmission electron microscopy (HADDF-STEM), elemental mapping, and energy dispersive x-ray spectroscopy (EDS) were performed by a field emission transmission electron microscope (FEI Talos F200X). X-ray diffraction (XRD) patterns of samples were taken using PAN alytical X'Pert PRO diffractometer with a Cu Kα (0.15418 nm) line, operating at 40 kV and 40 mA, with a PIXcel Medipix2 detector. The luminescence spectra and decay profiles were collected by a steady-state and transient spectrofluorometer (FLS 1000, Edinburgh Instruments Ltd) under continuous-wave laser excitation at 804 nm or 980 nm (Changchun New Industries Optoelectronics Technology Co., Ltd). All measured luminescence spectra have been calibrated by the spectral response of the instrument. Absorption spectra were acquired by a UV-Vis-NIR spectrophotometer (Cary 5000, Agilent Technologies Co., Ltd). Nanosecond transient absorption spectra (ns-TA) were obtained by a transient absorption spectrometer (EOS, Ultrafast Systems). Briefly, a femtosecond laser source (Astrella, Coherent) generates a pump beam (750 nm, 35 fs, 1 kHz) that passes an optical parameter amplifier (TOPAS, Spectra-physics) in order to tune the wavelength of fs laser. An ultra-continuum picosecond laser (2 kHz) was used as the probe beam. The probe beam passed a cuvette with 2 mm optical path and was collected by a high-speed spectrometer. The delay time (in nanosecond range) between pump and probe laser was set electronically, and analyzed by a time interval analyzer (CNT-90, Pendulum Instruments). Data were acquired automatically using a software based on LabView.

### Calculation of the upconversion luminescence enhancement fold

Following dye sensitization, the upconversion luminescence intensity of the perovskite $CsPbBr_3$:$Yb^{3+}$ nanocrystals increased by ~27,502-fold. This enhancement factor was calculated by comparing the integrated area of the upconversion luminescence spectra of the dye-sensitized $CsPbBr_3$:$Yb^{3+}$ nanocrystals (49,010,689) under 808 nm laser irradiation (8.4 $W/cm^2$), to that of non-sensitized $CsPbBr_3$:$Yb^{3+}$ nanocrystals (integrated area 1782) under 980 nm laser irradiation (8.4 $W/cm^2$) (Fig. S8 and Fig. S9).

### Measurement of the density IR783 in perovskite nanocrystals

**Calculation of the molar concentration of IR783 dye molecules.** According to the Lambert-Beer law, the absorbance of IR783 follows the equation:

$$\alpha = \varepsilon bc \qquad (2)$$

where $\alpha$ is the absorbance, $\alpha$ is the molar extinction coefficient, $b$ is the optical distance, and $c$ is the concentration of IR783. Since the $\varepsilon$ and $b$ are constants, Eq. (2) can be rewritten as:

$$\alpha = kc \qquad (3)$$

The constant $k$ can be determined by varying the concentration of IR783 and measuring the corresponding absorbance. To minimize the effect of reabsorption, the IR783 concentrations were kept low enough to ensure that the absorbance data aligned well with a straight line. As shown in Supplementary Fig. 8, k was estimated to be 349.71 (at 795 nm). For IR783 dye concentration coordinated on nanocrystal surface in Fig. 2c, a molar concentration of $2.07 \times 10^{-6}$ mmol/mL was calculated.

## Calculation of the molar concentration of CsPbBr$_3$:Yb$^{3+}$ nanocrystals

The absorbance of perovskite nanocrystals should also follow the Lambert-Beer law, as expressed in Eq. (2). Given that the doping concentration of Yb$^{3+}$ is extremely low, we assume that the ion doping does not affect the molar extinction coefficient. This coefficient can be estimated using the previously reported equation:

$$\varepsilon = 1.98 \times 10^{-2} \times d^3 \, cm^{-1} \, \mu M^{-1} \tag{4}$$

where $d$ denotes the diameter of nanocrystals (in nm). Based on this formula, the molar extinction coefficient of CsPbBr$_3$:Yb$^{3+}$ nanocrystals at 400 nm was calculated as 3867.19 cm$^{-1}$ μM$^{-1}$. Since the optical path used in the measurement was 1 cm, the molar concentration of CsPbBr$_3$:Yb$^{3+}$ nanocrystals was determined to be 5.01 × 10$^{-8}$ mmol/mL.

## Calculation of the number dye molecules per single CsPbBr$_3$:Yb$^{3+}$ nanocrystal

The density of IR783 molecules per nanocrystal was estimated as following:

$$D \frac{c_{IR-783}}{C_{perovskite \, nanocrystals}} \tag{5}$$

where c denotes the molar concentrations of IR783 and CsPbBr$_3$:Yb$^{3+}$ nanocrystals, respectively. For Fig. 2c with optimized dye concentrations, $D$ was calculated as 41.

## Calculation of the triplet energy transfer (TET) efficiency

the following equations are used to calculate the TET efficiency:

$$k_{undoped} = k_S + k_T \tag{6}$$

$$k_{doped} = k_S + k_T + k_{TET} \tag{7}$$

where $k_{undoped}$, and $k_{doped}$ refer to the decay rate of undoped NC-dye and doped NC-dye system. $k_S$, $k_T$ and $k_{TET}$ are denoted as the decay rate of singlet, triplet and TET rate, respectively. As a result, the TET rate can be extracted by:

$$k_{TET} = k_{doped} - k_{undoped} = \frac{1}{\tau_{doped}} - \frac{1}{\tau_{undoped}} \tag{8}$$

The efficiency of TET can be calculated by:

$$\eta_{TET} = \frac{k_{TET}}{k_S + k_T + k_{TET}} = \frac{k_{TET}}{k_{doped}} \tag{9}$$

## Reporting summary

Further information on research design is available in the Nature Portfolio Reporting Summary linked to this article.

## Data availability

All the relevant data that support the findings of this work are available from the corresponding author upon request. Source data are provided with this paper.

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

## Acknowledgements

This work was supported by the Key Technology Research and Industrialization Demonstration Project of Qingdao (Grant No. 25-1-1-gjgg-1-gx), the Outstanding Young Scholars Project of the Natural Science Foundation of Heilongjiang Province, China (Grant No. JJ2023JQ0025), the Opening Project of State Key Laboratory of Space Power Sources (Grant No. YF07050123F2531), the grants from the National Natural Science Foundation of China (Grant No. 51972084, Grant No. 52272270), the Young Scientist Workshop (Harbin Institute of Technology) (Grant No. AUGA5710094420), and the Fundamental Research Funds for the Central Universities, China (Grant No. AUGA5710052614, Grant No. HIT.OCEF.2023041).

## Author contributions

G.C. conceived the research; Y.Z. prepared the perovskite nanocrystals; Y. Z. collected most of the experimental data, with contributions from T.Z., Y.D., X.L., X.Z. and H.A.; Y.Z. and G.C. discussed and interpreted the collected data; all authors contributed to the data analysis; the manuscript was written by Y. Z., H.A. and G.C.; G.C. directed the research.

## Funding

## Competing interests

The authors declare no competing interests.
