## [Transparent Peer Review file · Nature Communications]

Near Infrared Sensitized Exciton Upconversion Luminescence from Inorganic Perovskite Nanocrystals

Corresponding Author: Professor Guanying Chen

Version 0:

Reviewer comments:

Reviewer #1

(Remarks to the Author)

In this manuscript, the author extended the responsive wavelength of inorganic lead halide perovskite nanocrystals into the near-infrared region by designing a coupled dye-perovskite nanocrystal system. This approach harnesses highly efficient triplet energy transfer between Yb³⁺ ions incorporated into the lead halide perovskite lattice and IR783 antenna molecules chemically coordinated on the nanocrystal surface. This system demonstrated robust, cooperative, sensitized upconversion luminescence of band-edge excitons, with absorption spanning a broad wavelength range from 600 to 860 nm. Dye sensitization achieved an over 27,500-fold enhancement in exciton upconversion luminescence intensity under continuous-wave laser irradiation at 804 nm (sub-10 Wcm⁻²), yielding an upconversion brightness of 3.22 M⁻¹cm⁻¹. Moreover, the article achieved precise multicolor upconverted exciton luminescence by tuning the bandgap of lead halide perovskite nanocrystals via halide mixing. This article presented a relatively novel way; however, there are still issues and characterizations that require further clarification and elaboration. It is recommended that these revisions be addressed prior to publication.

1. Environmental Stability Studies: It is recommended to investigate the long-term environmental stability of these upconverting nanocrystals under various conditions such as ambient air, water, and humidity. Testing stability under infrared laser irradiation over prolonged periods would also be valuable, as this is critical for practical applications, especially in bioimaging and optoelectronic devices.
2. Performance Benchmarking: Comparing the upconversion performance of IR783-sensitized CsPbBr₃:Yb³⁺ nanocrystals with other reported upconverting perovskite nanocrystals would enhance the impact of this work. Direct benchmarking in terms of quantum yield, brightness, and power dependence could provide a clearer picture of the advantages and limitations of this dye-sensitized system.
3. Extension of Upconversion Emission Wavelengths: To further broaden the applicability of the nanocrystals, we suggest modifying the halide composition to expand the upconversion emission into the blue wavelength range. This would enhance their utility in fields requiring multicolor imaging or specific wavelength emissions, such as high-resolution bioimaging and color-specific photodetection.
4. The argument that the difference in absorption spectra in different solvents proves the coordination between the sulfonic group of dye IR783 and the nanocrystals is somewhat weak. Please include additional tests, such as XPS or infrared spectroscopy, to directly demonstrate the change in the coordination environment.
5. What are the advantages of this material in the industry? Please add a table or graph to highlight the superior performance of this material.

Reviewer #2

(Remarks to the Author)

The manuscript by Chen et al. reported a novel strategy to achieve bright multicolor upconversion luminescence (UCL) from excitons in Yb³⁺-doped cesium lead halide perovskite nanocrystals (PeNCs) through organic dye-lanthanide tandem sensitization. The coordination of small sensitizing cyanine dyes (IR-783) to the surface of Yb³⁺-doped PeNCs entailed an intense and broad spectral response range of near-infrared (NIR) light from 600 nm to 860 nm. The collected spectroscopic and transient absorption experiments revealed an ultrafast triplet energy transfer process from the surface-anchored dyes to the lattice-incorporated Yb³⁺ ions with a near-unity efficiency, followed by a cooperative sensitization process that excited

the delocalized excitons. These results sound interesting and well supported by the experimental data. They are not only important for the development of sensitized upconversion system, but also a significant step forward to create NIR-responsive perovskites, which would probably excite interdisciplinary interests in the fields of solar cells, security, biosensing, and beyond. Therefore, I believe this manuscript is well-suited for publication in the journal of Nature Communications. Below are some minor suggestions for consideration of improvement:

1. Introduction: "nanoscale lead halide with a minimum bandgap of 1.73 eV". CsPbX₃ PeNCs with different halide compositions have different minimum bandgaps. Please clarify here and add a reference to support the claim.
2. What about the actual doping concentrations of Yb³⁺ in CsPbBr₃ PeNCs? The authors should elucidate it more clearly. ICP-AES measurements are suggested.
3. Why did the authors choose CsPbBr₃ PeNCs instead of CsPbCl₃ and CsPbI₃ for the investigation? Does the dye sensitization system work also for CsPbCl₃ and CsPbI₃?
4. Figure 3e. The authors filled the cuvette with different ratio of nitrogen and oxygen for the UCL measurements. Is it a volume or molar ratio? How to regulate the ratio? Please clarify it.
5. Usually, the dye sensitization system has the problem of photobleaching due to the instability of organic dyes. How about the proposed system?

Reviewer #3

(Remarks to the Author)

I co-reviewed this manuscript with one of the reviewers who provided the listed reports.

Reviewer #4

(Remarks to the Author)

This work describes the synthesis and photophysical investigations of novel upconverting materials consisting of lanthanide (Yb) doped lead-halide perovskite nanocrystals that are sensitized with NIR dye molecules. The research addresses a key challenge in this material space, adding NIR excitation functionality to perovskite nanocrystals (NCs), which are known to have exceptional optical properties, but primarily in the visible. Researchers have previously attempted to address this challenge by incorporating lanthanide ions into perovskite NCs (as is done here), which can upconvert IR excitation to energies that then pump the perovskite excitons. However, this previous scheme is flawed in that it trades one of the main advantages of the perovskites – a large absorption cross-section – for the very small absorption cross-sections of the lanthanides in order to achieve the NIR response. Here, they overcome this major tradeoff by sensitizing with NIR-active dyes with relatively large absorption cross-sections that effectively act as molecular antennas for the upconverting NCs. The authors show impressive results, fortified by key mechanistic studies and important controls that elucidate the underlying photophysics at the heart of the observed behavior. I particularly like the control experiment utilizing dye molecules without sulfonic groups to establish the dye attachment's chemical coordination to the NP surface, as well as the N₂-to-O₂ ratio response to establish dye-triplet contributions. The authors finish by showing results for dye sensitization on different perovskite compositions that emit over a range of visible wavelengths. Overall, I find the work to be of high quality and clearly explained. That said, I do have some comments, questions, and suggestions that should be addressed before recommending publication:

- 1) I don't understand the >100% (up to 200%) Yb doping levels listed in Fig. S5? Is this due to the different charge states of Yb vs. Pb? Please explain this better in the text.
- 2) Also in Fig. S5, the authors only show Yb doping levels of 40% and greater, with the 40% level giving the highest upconverted emission intensity. The authors should expand this plot with additional measurements showing emission spectra for lower Yb doping levels. For example 10 and 20% doping as well, to prove this is the optimal (or perhaps establish a better doping level!).
- 3) I commend the authors for quantifying the quantum yield, a key metric for these types of materials. Because it is an intensity-dependent value, the authors should also include the excitation intensity used for this measurement in the main text. Also: do the authors see any paths forward for improving the QY in the future? Perhaps this can be discussed in the conclusion or end discussion.
- 4) I don't actually see the yellow curve in 2d, though it is listed in the legend.
- 5) What do the authors mean by "low light excitation" in line 253? Please give the specific intensity. I know "sub-10W" is listed in the abstract and conclusion, but readers would benefit from the actual number being listed here.
- 6) The authors do an excellent job of showing how oxygen quenches the triplet behavior. This also brings up questions of stability under ambient conditions. The major drawback of these dye-sensitized systems is stability. The authors should add a discussion of stability and how the issue could be potentially alleviated.

A couple other small things:

- 7) Line 37, second sentence of the introduction: I believe the word "lost" should be "cost"?
- 8) Line 79: the beginning of the sentence should be rewritten; it should be either "proof of concept" or "demonstrate", but not both.

Reviewer #5

(Remarks to the Author)

In this communication, Zhang et al. report on an upconversion luminescence platform using Yb³⁺-doped halide perovskite

nanocrystals with infrared sensitizing dyes on the surface. They synthesize Yb³⁺-doped CsPb(Br,I)₃ nanocrystals, adsorb IR-783 dye to the surface, and observe photoluminescence (PL) at ~520 nm upon illumination at 804 nm. They characterize the photophysics of this process using a combination of synthesis and transient spectroscopies, concluding that upconversion proceeds through excitation of an IR-783 singlet, intersystem crossing to a triplet state, energy transfer to Yb³⁺ dimers, and upconversion to produce an emissive CsPbBr₃ exciton.

This upconversion demonstration occurs at relatively low irradiances and is an exciting compliment to the current literature, wherein halide perovskites commonly serve as a photoabsorber and single-to-triplet sensitizer. In comparison, the authors propose that Yb³⁺-CsPbBr₃ serves both as the sensitizer and the emitter. While the demonstration and proposal of lanthanide-to-semiconductor upconversion is exciting, the photophysical explanation requires further evidence and/or clarification. I believe that, if proven, the results will be of significant interest to the Nature Communication readership, and I recommend publication if the authors can address the follow major points.

The authors propose that energy transfers from IR-783 to Yb³⁺ dimers, followed by concerted upconversion to form a CsPbBr₃ exciton. While I am convinced that energy absorbed by the IR-783 dye emits at 520 and involves triplets, I do not find sufficient evidence that this occurs solely via Yb³⁺ sensitization. I detail a few major concerns about this point here as well as potential approaches for the authors to address them:

- Two critical pieces of evidence for Yb³⁺ facilitated energy transfer are 1) that Yb³⁺-doped CsPbBr₃ exhibits upconversion while undoped CsPbBr₃ does not and 2) that the IR738 triplet state decays rapidly in Yb³⁺-doped samples and is longer lived than in undoped samples. The first point is strong evidence. However, the excitation conditions are not described (wavelength and irradiance). These should be added to facilitate review.

- However, the evidence for the second point is not currently clear in the data. The transient kinetic traces in Fig. 4c are for different features and over different time ranges, seemingly comparing the lifetime of a singlet state to the lifetime of a triplet state. From the surface plots in Fig. 4a and b, there appears to be a feature, but it's unclear what this is. Additionally, the lack of experimental details precludes effective review.

- o The only speciation of the singlet vs triplet states is done via kinetic measurements, which is insufficient given the odd transient absorption behavior, which appears to rise and fall (Fig. S13). The authors could address this by including the spectral traces at different times and comparing these with transient absorption data of IR783 dyes. However, currently, the only photophysical evidence for Yb being involved in the energy transfer is a weak TA signal with surprising time kinetics at an unknown experimental condition.

- o The data in panel c need to be plotted over the same time range to understand if the authors' claim is valid.

- o Y-axes in Fig 4c are required. These obscure the very different time ranges.

- o The wavelength of the kinetic traces, fluences, and information about optical density must be included.

- Another broad issue is that the photophysics data focus on the dynamics of energy in IR783, rather than the energy arriving to Yb³⁺ or CsPbBr₃. This hampers the authors' claim that energy transfer to Yb³⁺ and then to CsPbBr₃. Indeed, there is no evidence that energy exists in Yb³⁺ or about when it arrives in CsPbBr₃. Specifically, it is strange that the transient absorption data in Figure 4 does not show the presence of an exciton in CsPbBr₃, which should occur at ~ 500 nm. These features are potentially obscured by the positive differential absorption feature of the IR783 dye. However, I would expect these to be present given the high optical density of CsPbBr₃ or in transient absorption spectra of CsPb(Br,I)₃ alloys. The authors should address where this signal is. The arrival of energy to Yb³⁺ and/or CsPbBr₃ is critical to establishing the role of Yb³⁺.

- o For example, one could propose an alternative mechanism wherein triplet-triplet annihilation (with ISC perhaps facilitated by surface Yb³⁺) directly excites an energy level with sufficient energy (~ 500 nm) to transfer to CsPbBr₃. This anti-Stokes PL has been observed in CsPbBr₃ previously. The authors should address if such a mechanism could be active.

- o Some potential experiments to prove that CsPbBr₃ is excited via Yb³⁺ include:

- o Time-resolved PL showing that the rise time of CsPbBr₃ PL occurs over a time scale corresponding to the observed 1 ns triplet decay, followed by transfer to Yb³⁺. For example, [Zhou ... Song et al. Nano Lett. 19 (2019)]. Here, Figure 4g shows rise time associate with energy transfer between lanthanides. One would expect a similar observation here from IR-783 to Yb³⁺ and from Yb³⁺ to CsPbBr₃.

- o Observation of Yb³⁺ PL at low irradiances after excitation of the IR-783 dye at ~ 804 nm.

- o Observation of that on a log-log plot of Yb³⁺ PL vs irradiance Yb³⁺ PL intensity decreases (relatively) at higher irradiances. A kinetic analysis for the rate of Yb³⁺ excitation

- In addition, the authors also claim that TET occurs to surface Yb³⁺ dimer pairs. While previous literature shows that these McPherson pairs likely, it also shows that very few dimers are active for downconversion [Erickson et al J Phys Chem C 123 (2019)]. The authors vary Yb³⁺ to make this claim, finding a nominal optimum of 40% Yb (presumably relative to Pb). Other literature shows that increasing Yb³⁺ improves downconversion. However, the authors appear to exhibit the opposite, weakening their claim that Yb³⁺ dimer pairs upconvert light. Addressing this seeming-discrepancy would improve the manuscript.

- More broadly, Yb³⁺ upconversion occurs in many materials without the need for dimers. For example, upconversion could occur through the triplet annihilation mechanism (described above), energy transfer upconversion, or a cooperative mechanism as proposed by the authors [Rabouw et al. ACS Nano 12 (2018)]. While an appealing hypothesis, I find no evidence to suggest that dimers are involved.

• Two important aspects of the data I don't understand are how the authors calculate the 98.4% TET efficiency and why there is such a long rise time in the Figure 4 TA spectra, despite the short IRF (Fig. S15). The first is particularly important and not clear to me. I suggest the authors address these points to aid non-expert readers.

The authors should also address the following minor issues:

- A common challenge in lanthanide-based upconversion literature is rare earth contamination. Indeed, the 976 nm PL spectrum in Fig 3c looks like it may include some Er³⁺ or Tb³⁺ at 550 nm. The authors should comment about potential contamination.
- The CsPbBr₃ PL is inconsistent throughout the manuscript. For example, the PL in Figure 3c and Figure 3E are remarkably different. The N₂:O₂=1:0 data are intensely structured. Given the potential influence of rare earth contamination, this should be discussed.
- Additional photophysics characterization of the Yb³⁺-doped and undoped CsPbBr₃ would improve the paper. Given evidence for lanthanide-based defects states in the CsPbBr₃ band gap [Milstein et al. Chem Mater 34 (2022)], you would expect to see a decrease in PL lifetime upon doping. I wonder how these defects, if present, may influence the upconversion process.
- The Yb³⁺ absorption spectrum in Figure 3D is very blue. It's extremely unusual for the Yb³⁺ peak to be blue of 1000 nm. The authors should also comment on this point to explain its energy compared to literature.
- The singlet lifetime analysis appears to be inconsistent. Figure S16 shows that the singlet is excited over ~ 4 ns (after ps excitation) and then decays from ~ 4 ns; while Figure S15 instead shows that the singlet is excited (after fs excitation) within 1 ps and decays by 1 ns. There may be a time-zero offset problem? Or the ps laser and detector have a long time constant? Plotting the instrument response function on the Figure S16 could resolve this. In addition, it isn't clear to me why the S15 transient absorption data lack a rise time, while the Figure 3a and b, Figure S14, and Figure S13 data do not have a rise time. Is this a feature of intersystem crossing?
- The authors claim the dye the ITO surface exhibits a 5-7 nm shift in optical absorbance due to solvent effects. Other authors have ascribed similar shifts to surface adsorption [Zou et al Nat Photon. 6 (2012)]. This may be additional evidence dyes are adsorbed onto the particle surface.
- The authors conduct a nice control experiment showing the importance of sulfonic acid groups on sensitizing dye molecules. However, it's unclear if these dyes adsorb on the nanocrystal surface without the sulfonic acid groups. Without confirmation that the dye is on the nanocrystal surface, it's impossible to assess this experiment. The authors should prove that this alternative dye is on the nanocrystal surface. Optical absorption spectra the CsPbBr₃-Yb³⁺-IR780 conjugate could confirm this, for example.
- Throughout the manuscript, there are a range of potential typos in figures and captions that make review and interpretation challenging. These must be fixed. Some particularly relevant ones include:
 - o Figure 2d – what are the y-axis units? Are these spectra comparable? What is the illumination wavelength and irradiance for CsPbBr₃-Yb³⁺? What sample is being studied – is this the 40% Yb³⁺?
 - o Figure 3e – what are the y-axis units?
 - o Figure 4c – the y-axis units should be included. Leaving these out obscures the data.
 - o Figure S3 – what nominal doping concentration is this data for?
- The quantum yields listed in Table S1 compare two-photon upconversion with dyes. However, the dyes are uncited and not representative of much more common dyes, such as Rhodamines, with photoluminescence quantum yields of ~ 50%. The authors should include more representative dyes in this table as well as references.

Version 1:

Reviewer comments:

Reviewer #1

(Remarks to the Author)

The quality of the revised manuscript has been improved and I recommend accept it.

Reviewer #2

(Remarks to the Author)

The article, in its present form, is much improved with respect to its previous edition. All the questions I raised have been well addressed or explained. I recommend it for publication without further revision.

Reviewer #3

(Remarks to the Author)

The article in its present form is much improved with respect to its previous edition. All my concerns have been well addressed or explained. I recommend it for publication without further change.

Reviewer #4

(Remarks to the Author)

I have gone over the updated files and response letter. I find that the authors have thoughtfully considered and addressed (whenever possible) the comments and questions posed by the reviewers.

My concerns were particularly focused on issues of stability. The authors have added new stability measurements that provide insight, and have updated the text on this front. I also appreciate their further attempts and control measurements for untangling the roll of Yb in the energy transfer and upconversion process.

I support publication of the updated manuscript in Nature Communications.

Reviewer #5

(Remarks to the Author)

This revised communication from Zhang et al. is significantly improved in revision. The upconversion demonstration is impressive, occurs at relatively low irradiances and is an exciting compliment to the current literature, wherein halide perovskites commonly serve as a photoabsorber and single-to-triplet sensitizer. However, the photophysics are still missing some critical pieces of data needed to for readers to understand the results. These are already collected. Provided that plotting these and adding relevant discussion does not change the results, I strongly recommend publication.

Photophysics control data:

In the authors' model, they propose that the IR-783 dye absorbs via the singlet state, undergoes intersystem crossing to form a triplet, which then transfer to Yb³⁺ in the CsPbX₃ nanocrystals. Here two excited Yb³⁺ atoms upconvert the energy forming an exciton. The photophysics in Figure 4A and S15 show that, without Yb³⁺ dopants, the IR-783 dye is excited and undergoes intersystem crossing to an excited triplet state. The singlet state is depopulated over ~ 3 ns. The photophysics in Figure 4B and S16 show that, with Yb³⁺ atoms, the IR-783 dye is excited, and the singlet state is depopulated over ~ 3 ns. The longer time region, during which the triplet state is presumably absent, is not shown. While the triplet state appears to be absent in the surface plot of Figure 4B, the optical densities are different. Indeed, there is a discrepancy in optical density between Figure 4 and SI Figures S15 and S16, where the OD in the manuscript is half the OD in the SI, highlighting this importance.

To their credit, the authors added much of this requested data and noted that it was challenging to measure the triplet state in Yb-doped samples. I sympathize with this. The authors should still add this control data as detailed below for readers:

- The authors should add the TA spectra at these longer times (~100-180 ns) for IR783-Yb³⁺-CsPbBr₃ samples, even if it shows nothing. In other words, Figure S15B should be reproduced for the IR783-Yb³⁺-CsPbBr₃ samples. This should be included to allow reviewers to assess the authors' claims. To their credit, the authors do note that the triplet dynamics were challenging to observe in the Yb-doped sample.
- In addition, the authors claim that the triplet state lifetime in IR783-Yb-CsPbBr₃ is 1 ns, but the TA spectra in the SI (Figure S15) appear to show only singlet character. Because, as the authors have added, the triplet state has a much lower optical density, it is likely that this is challenging or impossible to observe. The spectra in Figure S16 (and kinetic trace in Figure 4C) only show the decay of the singlet state, which the authors claim to be the decay of the triplet. This decay rate is the same as in Yb-CsPbBr₃. One explanation could be that the triplet state transfers at a rate much faster than the IRF. Are the authors measuring the singlet or triplet decay in this analysis? The authors should explain.
- This question is significant because this parameter is used to calculate the TET efficiency and rate. Would an IRF-limited triplet decay process change the triplet energy transfer efficiency calculation? The transfer rate may be much faster than reported.
- Likewise, Figure 4C still compares two entirely different features: the singlet decay of IR783-Yb-CsPbBr₃ (maybe this is also the triplet decay per the previous comment) vs. the triplet decay of IR783- CsPbBr₃. The IR783-Yb-CsPbBr₃ may contain no triplet signal at long times because it transfers. But this isn't shown in the data. In fact, this is obscured by normalizing the data in Figure 4C to different times.

The authors should include the full kinetic traces at 520 nm for both IR783-Yb-CsPbBr₃ and IR783-CsPbBr₃ over the same time range so that readers can compare in the SI. The authors added all data but these in this revision.

- Why are the transient absorption optical densities different in the SI vs. the manuscript?
- Why is the TA signal of the Yb-containing sample lower than the samples without Yb?

Role of Yb-dimers

• Finally, the authors propose that Yb dimers are the species responsible for upconversion. In their rebuttal, the authors attempted upconversion in IRF783-Yb-CsPbCl₃, which did not work. They claim that this demonstrates dimers must be active. However, the energy of two Yb³⁺ excitations (~ 2 eV) is less than the CsPbCl₃ band gap (~ 3 eV). While I appreciate the additional experiments, this upconversion process is not active in CsPbCl₃ because upconversion would require three Yb atoms. Thus, this experiment does not prove Yb-dimers are or aren't active.

It may indeed be the case that Yb dimers are important, but the authors do not show that dimers are necessary or involved. However, the authors do demonstrate that surface Yb is important for dye binding. These surface Yb species cannot form dimers. This discrepancy suggests that dimers may not be active.

This comment isn't to say that Yb dimers are or aren't involved. They may be. As written, the manuscript claims that dimers are involved without evidence. This could be addressed by adding new data showing that Yb dimer species are involved or by adjusting the manuscript to emphasize that upconversion could be enhanced by Yb dimers.

As a minor comment, Figure S5 caption has a typo and should be CsPbBr₃ :Yb³⁺ for the top panel.

Version 2:

Reviewer comments:

Reviewer #5

(Remarks to the Author)

The revised manuscript includes the requested essential data as well as new data. This version addresses all my comments, and I recommend it for publication.

RESPONS TO REVIEWER COMMENTS

Reviewer #1 (Remarks to the Author):

In this manuscript, the author extended the responsive wavelength of inorganic lead halide perovskite nanocrystals into the near-infrared region by designing a coupled dye-perovskite nanocrystal system. This approach harnesses highly efficient triplet energy transfer between Yb^{3+} ions incorporated into the lead halide perovskite lattice and IR783 antenna molecules chemically coordinated on the nanocrystal surface. This system demonstrated robust, cooperative, sensitized upconversion luminescence of band-edge excitons, with absorption spanning a broad wavelength range from 600 to 860 nm. Dye sensitization achieved an over 27,500-fold enhancement in exciton upconversion luminescence intensity under continuous-wave laser irradiation at 804 nm (sub-10 Wcm^{-2}), yielding an upconversion brightness of $3.22 \text{ M}^{-1}\text{cm}^{-1}$. Moreover, the article achieved precise multicolor upconverted exciton luminescence by tuning the bandgap of lead halide perovskite nanocrystals via halide mixing. This article presented a relatively novel way; however, there are still issues and characterizations that require further clarification and elaboration. It is recommended that these revisions be addressed prior to publication.

Response: We appreciate the positive comments and careful review on our work. In the revised manuscript, we have evaluated the stability of the upconverting nanocrystals under different conditions. A comparison table of the brightness, quantum yields as well as power density was added. Also, FTIR was measured giving the results that indicate the coordination of dye molecules and PeNCs. We hope our modifications have made this work better.

1. Environmental Stability Studies: It is recommended to investigate the long-term environmental stability of these upconverting nanocrystals under various conditions such as ambient air, water, and humidity. Testing stability under infrared laser irradiation over prolonged periods would also be valuable, as this is critical for practical applications, especially in bioimaging and optoelectronic devices.

Response: We thank the reviewer for careful review and insight. We added several stability measurements of the upconverting nanocrystals, and tested the long-term performance of these materials under daylight (**Figure R1.1**), infrared laser (**Figure R1.2**) and exposed to air (**Figure R1.3**).

We found that the stability of the system needs further improvement especially when exposed to the air. However, we also found the deterioration rate of the upconversion luminescence became slower after a period of exposure.

Figure R1.1. The upconversion intensity (a) and its tendency (b) of IR783-CsPbBr₃:Yb³⁺ under daylight radiation (excited by 804 nm laser, power density= 8.4 W/cm²).

Figure R1.2. The upconversion intensity (a) and its tendency (b) of IR783-CsPbBr₃:Yb³⁺ under 804 nm laser radiation (power density= 8.4 W cm⁻²).

Figure R1.3. The upconversion intensity (a) and its tendency (b) of IR783-CsPbBr₃:Yb³⁺ after exposed to ambient atmosphere (excited by 804 nm laser, power density= 8.4 W/cm²).

Action: We have added the results from stability measurements under three different conditions to the supplementary information. And we added text *“The stability of this system under different conditions was further evaluated (Supplementary Fig. S22-24). Although the stability of the materials needs further improvement,”* In the conclusion.

2. Performance Benchmarking: Comparing the upconversion performance of IR783-sensitized CsPbBr₃:Yb³⁺ nanocrystals with other reported upconverting perovskite nanocrystals would enhance the impact of this work. Direct benchmarking in terms of quantum yield, brightness, and power dependence could provide a clearer picture of the advantages and limitations of this dye-sensitized system.

Response: We appreciate the suggestion. In the revised manuscript, a comparison table was added (Table R1.1) in supporting information which summarizes the brightness, quantum yields and power density of different upconverting nanocrystals and different dyes.

TableR1.1. Comparison of achieved brightness along with their extinction coefficients, quantum yields and the power density with literature-reported well-established dye molecules and upconverting nanocrystals.

	Ex/Em(nm)	$\epsilon(\text{M}^{-1} \text{cm}^{-1})$	QY(%)	B ($\text{M}^{-1} \text{cm}^{-1}$)	PD(W/cm^2)
NCs/IR-783	808/515	261000	0.0012	3.22	8.4
LiYbF ₄ :0.005Tm ³⁺ @LiYF ₄ -CsPbBr ₃	980/430-580	220000	0.39	858	100
NaYF ₄ :0.2Yb ³⁺ ,0.02Er ³⁺	808/510-570	2880000	0.0001	2.88	533
NaYF ₄ : 0.15Yb ³⁺ , 0.2Er ³⁺	970/510-680	4.545	0.11	0.005	181
NaYF ₄ : 0.2Yb ³⁺ , 0.02Er ³⁺	980/510-570	1000000	0.0038	38	1128
CH1055	750/1055	10000	0.03	3	downconversion
BTC1070	1014/1070	115000	0.016	18.40	downconversion
Er ³⁺ [Zn(II)MC _{quinHA}]	380/1530	55000	9.9*10 ⁻⁴	0.5445	downconversion
ICG	785/821	270000	2.4	5400	downconversion
Eu ³⁺ [Ligand] ³⁻	340/605-720	0.067	34	0.0228	downconversion
DM-Naph	441/554	1088889	0.009	9800	downconversion
Rho110	495/520	78000	88	68640	downconversion
Rho560	559/587	97000	12	11640	downconversion
Ag ₂ S	808/1200	460000	0.08	368	downconversion

Action: We have inserted the new table which evaluated these results in the revised manuscript (supplementary Table S2).

3. Extension of Upconversion Emission Wavelengths: To further broaden the applicability of the nanocrystals, we suggest modifying the halide composition to expand the upconversion emission into the blue wavelength range. This would enhance their utility in fields requiring multicolor imaging or specific wavelength emissions, such as high-resolution bioimaging and color-specific photodetection.

Response: We thank the reviewer for the suggestion. We have tried to extend the emission to the blue range by sensitizing Yb³⁺ doped CsPbCl₃ and CsPb(Cl/Br)₃ PeNCs. However, we found the dye sensitization system doesn't work in these systems. This may be because of that the bandgap of blue-emitting PeNCs is larger than the sum of two Yb³⁺ ions, causing ultralow cooperative-sensitized upconversion efficiency.

4. The argument that the difference in absorption spectra in different solvents proves the coordination between the sulfonic group of dye IR783 and the nanocrystals is somewhat weak. Please include additional tests, such as XPS or infrared spectroscopy, to directly demonstrate the change in the coordination environment.

Response: We agree with the comment. In the revised manuscript, we have added the results from infrared spectroscopy (**Figure R1.4**). As shown in the figure, features of S=O and O=S=O appear as well as of C-H in IR783-CsPbBr₃:Yb³⁺ complex, indicating the sulfonic acid group partially substitutes the oleic acid group.

Figure R1.4. Fourier transform infrared spectroscopy of CsPbCl₃:Yb³⁺ (top), IR783-CsPbBr₃:Yb³⁺ (middle) and IR783 (bottom).

Action: We have added the sentence “*The result from FTIR spectroscopy further proved the attachment (Supplementary Fig. S5)*” in the main text and the result from FTIR measurement was given in supplementary information.

5. What are the advantages of this material in the industry? Please add a table or graph to highlight the superior performance of this material.

Response: We thank the reviewer for the comment. The simple fabrication of this material and its low cost as well as the low excitation power density (sub-10 W/cm²) are the most important factors

in the industry. Besides, we hope the **Table R1.1** is able to demonstrates the superior performance of this material.

Reviewer #2 (Remarks to the Author):

The manuscript by Chen et al. reported a novel strategy to achieve bright multicolor upconversion luminescence (UCL) from excitons in Yb³⁺-doped cesium lead halide perovskite nanocrystals (PeNCs) through organic dye-lanthanide tandem sensitization. The coordination of small sensitizing cyanine dyes (IR-783) to the surface of Yb³⁺-doped PeNCs entailed an intense and broad spectral response range of near-infrared (NIR) light from 600 nm to 860 nm. The collected spectroscopic and transient absorption experiments revealed an ultrafast triplet energy transfer process from the surface-anchored dyes to the lattice-incorporated Yb³⁺ ions with a near-unity efficiency, followed by a cooperative sensitization process that excited the delocalized excitons. These results sound interesting and well supported by the experimental data. They are not only important for the development of sensitized upconversion system, but also a significant step forward to create NIR-responsive perovskites, which would probably excite interdisciplinary interests in the fields of solar cells, security, biosensing, and beyond. Therefore, I believe this manuscript is well-suited for publication in the journal of Nature Communications. Below are some minor suggestions for consideration of improvement:

We thank for the positive comments and careful review on our work. In the revised manuscript, we have evaluated the stability of the upconverting nanocrystals under different conditions. The actual Yb doping concentration was given through ICP-OES. Some writing mistakes have been fixed. We hope our modifications have made this work better.

1. Introduction: “nanoscale lead halide with a minimum bandgap of 1.73 eV”. CsPbX₃ PeNCs with different halide compositions have different minimum bandgaps. Please clarify here and add a reference to support the claim.

Response: We thank the reviewer for the comment. We agree with that the description might be perceived as ambiguous. In the revised manuscript, we added an explanation. Changes to the manuscript: “nanoscale lead halide with a minimum bandgap of 1.73 eV (when X= I)”

Action: We have specified the condition in Line 39.

2. What about the actual doping concentrations of Yb³⁺ in CsPbBr₃ PeNCs? The authors should elucidate it more clearly. ICP-AES measurements are suggested.

Response: We agree with the comment. In the revised manuscript, we added result from ICP-OES measurements (**Table R2.1**).

Table R2.1. Yb³⁺ doping concentration in PeNCs

sample	Yb/Pb feeding ratio	Nominal Yb doping concentration (mol%)	Actual Yb doping concentration from ICP-OES (mol%)
CsPbBr ₃	undoped	0.0	0.0
CsPbBr ₃ : 0.08Yb ³⁺	0.4:1	40	0.80
CsPbBr ₃ : 0.2Yb ³⁺	1:1	100	1.79
CsPbBr ₃ : 0.3Yb ³⁺	1.5:1	150	2.91
CsPbBr ₃ : 0.4Yb ³⁺	2:1	200	3.30

Action: We have added the text “*while the actual doping concentration is 0.8%*” after the nominal concentration and added the table to the revised manuscript (**supplementary Table S1**).

3. Why did the authors choose CsPbBr₃ PeNCs instead of CsPbCl₃ and CsPbI₃ for the investigation?

Does the dye sensitization system work also for CsPbCl₃ and CsPbI₃?

Response: We appreciate the comment. CsPbBr₃ PeNCs were chosen for the investigation mainly considering its high PL quantum yields which indicates efficient radiative relaxation and larger Stokes shift. The dye sensitization system works for CsPbI₃ but not for CsPbCl₃. That also explains why CsPbCl₃ PeNCs was dismissed as the sample. As we explained to reviewer1 (comment 3), we think this may be because of that the bandgap of blue-emitting PeNCs is larger than the sum of two Yb³⁺ ions, causing ultralow cooperative-sensitized upconversion efficiency. As for the CsPbI₃ PeNCs, the emission wavelength (~750 nm) is closer to our excitation wavelength (804 nm) as well as the wide absorption region of dye, causing complex and overlapping signals.

4. Figure 3e. The authors filled the cuvette with different ratio of nitrogen and oxygen for the UCL measurements. Is it a volume or molar ratio? How to regulate the ratio? Please clarify it.

Response: We appreciate the reviewer’s comment. It is the volume ratio achieved through two flowmeters which connect to a nitrogen cylinder and an oxygen cylinder, respectively, also connecting to the same cuvette. By tuning the flow rate, we assume the gas can be mixed as setting.

5. Usually, the dye sensitization system has the problem of photobleaching due to the instability of organic dyes. How about the proposed system?

Response: We thank the reviewer for the constructive suggestion. We added several stability measurements of the upconverting nanocrystals, testing the long-term performance of these materials under daylight, infrared laser and exposed to air (referred to Reviewer1, comment 1, Figure R1.1, R1.2, R1.3). The stability of the system needs further improvements especially when exposed to the air. However, we also found the deterioration rate of the upconversion luminescence became slower after a period of exposure.

Action: We have added the results from stability measurements under three different conditions to supplementary information. And we added text “*The stability of this system under different conditions was further evaluated (Supplementary Fig. S22-24). Although the stability of the materials needs further improvement,*” in the conclusion.

Reviewer #3 (Remarks to the Author):

I co-reviewed this manuscript with one of the reviewers who provided the listed reports.

Response: We appreciate the reviewer for the careful review.

Reviewer #4 (Remarks to the Author):

This work describes the synthesis and photophysical investigations of novel upconverting materials consisting of lanthanide (Yb) doped lead-halide perovskite nanocrystals that are sensitized with NIR dye molecules. The research addresses a key challenge in this material space, adding NIR excitation functionality to perovskite nanocrystals (NCs), which are known to have exceptional optical properties, but primarily in the visible. Researchers have previously attempted to address this challenge by incorporating lanthanide ions into perovskite NCs (as is done here), which can upconvert IR excitation to energies that then pump the perovskite excitons. However, this previous scheme is flawed in that it trades one of the main advantages of the perovskites – a large absorption cross-section – for the very small absorption cross-sections of the lanthanides in order to achieve the NIR response. Here, they overcome this major tradeoff by sensitizing with NIR-active dyes with relatively large absorption cross-sections that effectively act as molecular antennas for the upconverting NCs. The authors show impressive results, fortified by key mechanistic studies and important controls that elucidate the underlying photophysics at the heart of the observed behavior. I particularly like the control experiment utilizing dye molecules without sulfonic groups to establish the dye attachment's chemical coordination to the NP surface, as well as the N₂-to-O₂ ratio response to establish dye-triplet contributions. The authors finish by showing results for dye sensitization on different perovskite compositions that emit over a range of visible wavelengths. Overall, I find the work to be of high quality and clearly explained. That said, I do have some comments, questions, and suggestions that should be addressed before recommending publication:

We appreciate for the positive comments and careful review on our work. In the revised manuscript, the actual Yb doping concentration was given through ICP-OES except the nominal concentration. The quantum yield of upconverting nanocrystal was given and the specific power density values have been added when needed. Also, we have fixed some writing mistakes you have mentioned (we really appreciate that). We hope our modifications have made this work better.

1) I don't understand the >100% (up to 200%) Yb doping levels listed in Fig. S5? Is this due to the different charge states of Yb vs. Pb? Please explain this better in the text.

Response: We appreciate the reviewer for the comment. The nominal concentration is calculated following the formula:

$$c(\text{Yb}^{3+}) = \frac{n(\text{Yb}^{3+})}{n(\text{Pb}^{2+})}$$

where c denotes the nominal concentration and n denotes the molar quantity of different ions. To eliminate further misunderstanding, we added a table (referred to Reviewer2, comment 2, Table R2.1) in the revised manuscript to explain the Yb^{3+} doping concentration.

Action: Line 138, we have added the text while the actual doping concentration is 0.8% after the nominal concentration in original manuscript and added the table to the supplementary information.

2) Also in Fig. S5, the authors only show Yb doping levels of 40% and greater, with the 40% level giving the highest upconverted emission intensity. The authors should expand this plot with additional measurements showing emission spectra for lower Yb doping levels. For example 10 and 20% doping as well, to prove this is the optimal (or perhaps establish a better doping level!).

Response: We appreciate the reviewer for the comment. We synthesized a series of Yb doped PeNCs to find the optimal concentration. However, when the nominal concentration is lower than 40%, the actual Yb^{3+} concentration (measured by ICP-OES) was too weak to confirm the existence of Yb^{3+} so we excluded the samples with lower Yb^{3+} doping concentration.

3) I commend the authors for quantifying the quantum yield, a key metric for these types of materials.

Response: We agree with that measuring the quantum yield of these materials will improve this work. In the revised manuscript, we added the quantum yield ($1.23 \times 10^{-3}\%$) of the material in the main text (Line 145). The low quantum yield seems uncommon considering the bright luminescence. There are two reasons for this result. On one hand, the absolute quantum yield was underestimated owing to the strong reabsorption between dye molecules. On the other hand, the absorption cross-section of IR783 is large (in our work, even larger optical density than PeNCs), achieving the high brightness despite a low quantum yield.

Action: We added the text in Line 145: “quantum yield of $1.23 \times 10^{-3}\%$ (excited at 804 nm, power density = 8.4 W/cm^2)”

4) Because it is an intensity-dependent value, the authors should also include the excitation intensity used for this measurement in the main text. Also: do the authors see any paths forward for improving the QY in the future? Perhaps this can be discussed in the conclusion or end discussion.

Response: We thank the reviewer for the suggestion. In the revised manuscript, we added the specific power density in the main text when needed (Line 145, Line 206, Line 266, Line 275). We think two strategies may help to further improve the QY. On one hand, modifying the dye molecules to decrease the distance between dye and Yb^{3+} ions may further accelerate the energy transfer pathway since the distance between donor and acceptor is of great importance. In this aspect, constructing a $\text{CsPbBr}_3@ \text{CsPbBr}_3:\text{Yb}^{3+}$ core shell PeNCs may also work. The other strategy is to design new upconversion pathways. As reported, the cooperative sensitized upconversion pathway is weak compared to other pathways in fluoride nanocrystals. This may be the same in PeNCs. Designing another pathway such as energy transfer upconversion may overcome this problem. This part has been added to the conclusion.

Action: We have added the text in Line 302: “The stability of this system under different conditions was further evaluated. Although the stability of the materials needs further improvement, the dye sensitizing strategy is ideal to accomplish bright upconversion luminescence of PeNCs. Also, the quantum yield may be further enhanced by modifying the dye molecule and optimizing the distance between dye and Yb^{3+} ”.

5) I don't actually see the yellow curve in 2d, though it is listed in the legend.

Response: Tanks - The yellow curve was overlapped by the red curve since both curves are extremely weak compared with the blue curve. In the revised manuscript, we change the red solid line to red dashed line as shown in **Figure R4.1**. Hope this will help reading.

Figure R4.1. Upconversion luminescence spectra of CsPbBr₃:Yb³⁺ nanocrystals (red dashed line), IR783-CsPbBr₃ nanocrystals (yellow solid line), IR783-CsPbBr₃:Yb³⁺ nanocrystals (blue solid line), in contrast to the Stokes luminescence spectrum of IR783-CsPbBr₃:Yb³⁺ nanocrystals under 365 nm excitation (blue dashed line). Dye-sensitized upconversion spectra from IR783-CsPbBr₃:Yb³⁺ and IR780-CsPbBr₃:Yb³⁺ nanocrystals were acquired under 804 nm continuous-wave laser irradiance at 8.4 W/cm², while upconversion spectra from CsPbBr₃:Yb³⁺ nanocrystals were acquired under 980 nm continuous-wave laser irradiance at 8.4 W/cm².

Action: We have changed the red solid curve in Fig. 2d to dashed curve.

- 6) What do the authors mean by "low light excitation" in line 253? Please give the specific intensity. I know "sub-10W" is listed in the abstract and conclusion, but readers would benefit from the actual number being listed here.

Response: We appreciate the reviewer for the helpful suggestion. We have added the specific power density of the laser.

Action: The sentence "with visualized upconversion color shifting from green to orange and then to red under low light excitation at 804 nm (Fig. 5d)" was changed to "with visualized upconversion color shifting from green to orange and then to red under 804 nm excitation (Fig. 5d, power density= 8.4 W/cm²)"

- 7) The authors do an excellent job of showing how oxygen quenches the triplet behavior. This also brings up questions of stability under ambient conditions. The major drawback of these dye-sensitized

systems is stability. The authors should add a discussion of stability and how the issue could be potentially alleviated.

Response: We thank the reviewer for the helpful comment. In the revised manuscript, we added several stability measurements of the upconverting nanocrystals, testing the long-term performance of these materials under daylight, infrared laser and exposed to air (referred to Reviewer1, comment 1, Figure R1.1, R1.2, R1.3). The stability of the system needs further improvements especially when exposed to the air. However, we also found the deterioration rate of the upconversion luminescence became slower after a period of exposure.

Action: We have added the results from stability measurements under three different conditions to supplementary information. And we added text “The stability of this system under different conditions was further evaluated (Supplementary Fig. S20-22). Although the stability of the materials needs further improvement,” In the conclusion.

A couple other small things:

7) Line 37, second sentence of the introduction: I believe the word “lost” should be “cost”?

Response: Thanks. It is a spelling mistake and we have fixed it.

Action: Line37, the word *lost* have been replaced by *cost*.

8) Line 79: the beginning of the sentence should be rewritten; it should be either “proof of concept” or “demonstrate”, but not both.

Response: Thanks... We have deleted the “proof of concept” to avoid the semantic repetition.

Action: Line 79, the words *proof of concept* has been deleted.

Reviewer #5 (Remarks to the Author):

In this communication, Zhang et al. report on an upconversion luminescence platform using Yb^{3+} -doped halide perovskite nanocrystals with infrared sensitizing dyes on the surface. They synthesize Yb^{3+} -doped $\text{CsPb}(\text{Br},\text{I})_3$ nanocrystals, adsorb IR-783 dye to the surface, and observe photoluminescence (PL) at ~ 520 nm upon illumination at 804 nm. They characterize the photophysics of this process using a combination of synthesis and transient spectroscopies, concluding that upconversion proceeds through excitation of an IR-783 singlet, intersystem crossing to a triplet state, energy transfer to Yb^{3+} dimers, and upconversion to produce an emissive CsPbBr_3 exciton.

This upconversion demonstration occurs at relatively low irradiances and is an exciting compliment to the current literature, wherein halide perovskites commonly serve as a photoabsorber and single-to-triplet sensitizer. In comparison, the authors propose that Yb^{3+} - CsPbBr_3 serves both as the sensitizer and the emitter. While the demonstration and proposal of lanthanide-to-semiconductor upconversion is exciting, the photophysical explanation requires further evidence and/or clarification. I believe that, if proven, the results will be of significant interest to the Nature Communication readership, and I recommend publication if the authors can address the follow major points.

We sincerely appreciate the reviewers' insightful comments and thorough evaluation of our work. We fully agree with the scientific perspectives raised in the suggestions, many of which align with the constructive feedback previously provided by other reviewers and have been duly addressed in this revision. Undoubtedly, these recommendations have substantially contributed to enhancing the quality of this manuscript. While certain experimental constraints limited the scope of additional validations, we have made every effort to incorporate additional experimental data and provided enhanced theoretical explanations to address the raised concerns. We trust that these revisions have significantly strengthened the scientific rigor and clarity of the presented research.

The authors propose that energy transfers from IR-783 to Yb^{3+} dimers, followed by concerted upconversion to form a CsPbBr_3 exciton. While I am convinced that energy absorbed by the IR-783 dye emits at 520 and involves triplets, I do not find sufficient evidence that this occurs solely via Yb^{3+} sensitization. I detail a few major concerns about this point here as well as potential approaches for the authors to address them:

- Two critical pieces of evidence for Yb³⁺ facilitated energy transfer are 1) that Yb³⁺-doped CsPbBr₃ exhibits upconversion while undoped CsPbBr₃ does not and 2) that the IR783 triplet state decays rapidly in Yb³⁺-doped samples and is longer lived than in undoped samples. The first point is strong evidence. However, the excitation conditions are not described (wavelength and irradiance). These should be added to facilitate review.

Response: We appreciate the reviewer for the comments. The excitation conditions for doped and undoped are completely the same (excited by 804 nm laser, power density= 8.4 W/cm²) and we have now explained this in the revised manuscript (Line 133).

Action: We have added the excitation conditions in Line 133 in the main text: *This fact verifies that the sensitized upconversion luminescence stems from delocalized excitons. In sharp contrast, no upconversion luminescence were observed in the IR783-CsPbBr₃ control group (excited by 804 nm laser, power density= 8.4 W/cm²), confirming the importance of Yb³⁺ dopants to induce exciton upconversion luminescence.*

- However, the evidence for the second point is not currently clear in the data. The transient kinetic traces in Fig. 4c are for different features and over different time ranges, seemingly comparing the lifetime of a singlet state to the lifetime of a triplet state. From the surface plots in Fig. 4a and b, there appears to be a feature, but it's unclear what this is. Additionally, the lack of experimental details precludes effective review.

Response: We appreciate your comment. The excite-state absorption dynamics of 522 nm presents both singlet and triplet dynamics of the IR783. The transient absorption spectrum of IR783-CsPbBr₃ nanocrystals contains the signals of singlet and triplet of IR783. The kinetics trace over full time ranges was given in Figure S13. In the first 10 ns, the rise and fall can be ascribed to the singlet state which is in accordance to the ground-state bleach (Figure S15) while the excited-state absorption from 50 ns to 1000 ns arises from the triplet state of IR783 considering the lifetime of 67.62 ± 1.37 ns which corresponds to other rhodamine dyes (*Journal of Photochemistry*, 9 (1978) 411-424). But in the presence of Yb³⁺, the singlet-intra system crossing-triplet process was strongly accelerated and the singlet signal was overlapped by the ultrafast triplet decay. As a result, the triplet states of IR783 form at different timescale in undoped and Yb³⁺ doped CsPbBr₃ nanocrystals. To visually demonstrate the acceleration of triplet kinetics, we have now extracted the signal of triplet states in undoped

sample in Figure 4c and given the overall kinetics in the supporting information (Fig S15-S18 in the revised SI file).

The only speciation of the singlet vs triplet states is done via kinetic measurements, which is insufficient given the odd transient absorption behavior, which appears to rise and fall (Fig. S13). The authors could address this by including the spectral traces at different times and comparing these with transient absorption data of IR783 dyes. However, currently, the only photophysical evidence for Yb being involved in the energy transfer is a weak TA signal with surprising time kinetics at an unknown experimental condition.

Response: We agree with the reviewer. Limited by the large overlap between the singlet and triplet signals over the wavelength, the singlet and the triplet spectra of IR783 were differentiated over timescale. As shown in the spectra, singlet state (Figure R5.1a) presented strong wide excited-state absorption peaked at 522 nm and 560 nm while triplet state (Figure R5.1b) absorbed intensely at 480 nm and 522 nm, in accordance with the results from reported paper (*Phys. Scr.* 94 (2019) 095501 (15pp)). In the case of IR783-CsPbBr₃: Yb³⁺, it became difficult to extract the singlet and triplet spectra owing to the accelerated dynamics. As shown in Fig. R5.2, strong absorption was observed at 522 nm along with relatively weak absorptions at 480 nm and 560 nm, which proved our expectation that the spectra were highly overlapped. Hence, a triplet lifetime was estimated in the main text to evaluate the triplet transfer efficiency. Despite the difficulties in fully interpreting the complex TA dynamics, the relationship between Yb³⁺ doping concentration and upconversion intensity (Fig S7) is believed to adequately demonstrate the intermediate role of Yb³⁺: for the undoped sample, photon upconversion is negligible. With doping concentration increasing, the upconversion intensity varies.

Figure R5.1. Transient absorption spectra of singlet (a) and triplet (b) of IR783 in IR783- CsPbBr₃.

Figure R5.2. Transient absorption spectra of IR783-CsPbBr₃: Yb³⁺.

Action: We have added the TA spectra (supplementary Fig. S15, S16) to the revised manuscript. Also, the explanation *“Limited by the large overlap between the singlet and triplet signals over the wavelength, the singlet and the triplet spectra of IR783 were differentiated over timescale. As shown in the spectra, singlet state (Fig. S15a) presented strong wide excited-state absorption peaked at 522 nm and 560 nm while triplet state (Fig. S15b) absorbed intensely at 480 nm and 522 nm, in accordance with the results from reported paper. In the case of IR783-CsPbBr₃: Yb³⁺, it became difficult to extract the singlet and triplet spectra owing to the accelerated dynamics.”* was added to the main text.

o The data in panel c need to be plotted over the same time range to understand if the authors' claim is valid.

Response: We thank the reviewer for the comment and nice suggestion. As we explained above, we think this kind of demonstration can help readers to understand the acceleration of triplet state in IR783 more visually.

o Y-axes in Fig 4c are required. These obscure the very different time ranges.

Response: Thanks. In the revised manuscript, the Y-axis along with units were added.

Action: We have added the Y-axis as well as the units *a.u.* of Fig. 4c.

o The wavelength of the kinetic traces, fluences, and information about optical density must be included.

Response: Thanks. The wavelength, pulse power, and the Y-axis (which is corresponding to the optical density) of the kinetic traces have been added in the revised manuscript. Other conditions of the transient absorption measurements have described in the supplementary information.

Action: Line 245, we have added the conditions “detected at 522 nm, fluence = 1.58 μJ/cm²”. The Y-axis of Fig. 4c have been added.

- Another broad issue is that the photophysics data focus on the dynamics of energy in IR783, rather than the energy arriving to Yb³⁺ or CsPbBr₃. This hampers the authors’ claim that energy transfer to Yb³⁺ and then to CsPbBr₃. Indeed, there is no evidence that energy exists in Yb³⁺ or about when it arrives in CsPbBr₃. Specifically, it is strange that the transient absorption data in Figure 4 does not show the presence of an exciton in CsPbBr₃, which should occur at ~ 500 nm. These features are potentially obscured by the positive differential absorption feature of the IR783 dye. However, I would expect these to be present given the high optical density of CsPbBr₃ or in transient absorption spectra of CsPb(Br,I)₃ alloys. The authors should address where this signal is. The arrival of energy to Yb³⁺ and/or CsPbBr₃ is critical to establishing the role of Yb³⁺.

o For example, one could propose an alternative mechanism wherein triplet-triplet annihilation (with ISC perhaps facilitated by surface Yb³⁺) directly excites an energy level with sufficient energy (~ 500 nm) to transfer to CsPbBr₃. This anti-Stokes PL has been observed in CsPbBr₃ previously. The authors should address if such a mechanism could be active.

Response: We appreciate your comments. We hope to see the dynamics of excitons as expected. We think the upconverting process can be described as a nonradiative energy cascade process similar as reported (*J. Am. Chem. Soc.* 2020, 142, 11270-11278). However, this hypothesis will need more evidence to prove and we think our focus of this paper should be dye sensitized cooperative upconverting. Here, the possibility of triplet-triplet annihilation is excluded: as reported before, triplet-triplet annihilation demands for an energy matching singlet state and we have not observed such a state in IR783 (*Chem. Rev.* 2021, 121, 9165-9195, *Chem. Eur. J.* 2009, 15, 9191–9200); secondly, the upconverting emissions of excitons can be observed in CsPb(Br,I)₃ but is not working

in CsPbCl₃ nanocrystals (excited by 804 nm laser), indicating the importance of Yb-dimers considering the bandgap alignment. Meanwhile, we think that the dependence between Yb³⁺ doping concentration and upconversion emission intensity shown in Fig S7 has adequately demonstrated the intermediate role in the upconversion process: for the undoped sample, photon upconversion is negligible. With doping concentration increasing, the upconversion intensity varies. Based on the discussion above, the mechanism we come up with seems to be the most reasonable one.

o Some potential experiments to prove that CsPbBr₃ is excited via Yb³⁺ include:

- Time-resolved PL showing that the rise time of CsPbBr₃ PL occurs over a time scale corresponding to the observed 1 ns triplet decay, followed by transfer to Yb³⁺. For example, [Zhou ... Song et al. Nano Lett. 19 (2019)]. Here, Figure 4g shows rise time associate with energy transfer between lanthanides. One would expect a similar observation here from IR-783 to Yb³⁺ and from Yb³⁺ to CsPbBr₃.
- Observation of Yb³⁺ PL at low irradiances after excitation of the IR-783 dye at ~ 804 nm.
- Observation of that on a log-log plot of Yb³⁺ PL vs irradiance Yb³⁺ PL intensity decreases (relatively) at higher irradiances.
- A kinetic analysis for the rate of Yb³⁺ excitation.

Response: We thank the reviewer for the valuable and constructive suggestions. After multiple attempts, we found it difficult to detect the weak Yb emission due to the strong emission tail of IR783. Meanwhile, limited by the temporal resolution of the instrument, the fast PL dynamics of cyanine dye (< 1 ns) is undiscernible. Some failed attempts are presented below (Figure R5.3). Despite the unsuccessful detection of Yb emission and PL dynamics, we think the dependence between upconversion intensity and Yb doping concentration is sufficient to prove the role of Yb.

Figure R5.3. NIR PL spectrum of IR783-CsPbBr₃: Yb³⁺.

- In addition, the authors also claim that TET occurs to surface Yb³⁺ dimer pairs. While previous literature shows that these McPherson pairs likely, it also shows that very few dimers are active for downconversion [Erickson et al J Phys Chem C 123 (2019)]. The authors vary Yb³⁺ to make this claim, finding a nominal optimum of 40% Yb (presumably relative to Pb). Other literature shows that increasing Yb³⁺ improves downconversion. However, the authors appear to exhibit the opposite, weakening their claim that Yb³⁺ dimer pairs upconvert light. Addressing this seeming-discrepancy would improve the manuscript.

Response: We thank the reviewer for the comments. As you noticed, we do observe the opposite tendency while increasing the Yb³⁺ doping concentration. To eliminate the probability of random errors, we repeated the series of NCs for three times and the results exclude the probability. We find the Yb acetate has problem to solve when the concentration increases in our synthesis, which may explain the decreasing upconversion in highly doped nanocrystals. Besides, our claim is that increasing Yb³⁺ decreases upconversion instead of downconversion (the PL spectra in Figure S5 are measured under 804 nm laser, power density= 8.4 W/cm²) and we have fixed the wrong labels of Figure S5 (**Supporting information, Line 137**). As for the dimer pairs, we agree with that the dimers are not active for downconversion while we do think the dimers commonly exist in perovskites. As reported by Gamelin et al (*DOI: 10.1103/PhysRevMaterials.6.025404* and *DOI: 10.1103/PhysRevMaterials.6.074601*) and Ågren et al (*J. Phys. Chem. Lett. 2019, 10, 487–492*), the

bent $[\text{Yb}_{\text{Pb}}\text{-V}_{\text{Pb}}\text{-Yb}_{\text{Pb}}]^0$ dimers are more likely to form in perovskites nanocrystals mainly because of the charge balance and low formation energy. While these dimers are less active in downconversion, we think they are active in upconversion process considering the different lifetime of excitons and Yb^{3+} (if they are inactive, the upconversion will be difficult to take place).

Action: We have replaced the text *CsPbBr₃:Yb³⁺ nanocrystals* to *IR783-CsPbBr₃:Yb³⁺ nanocrystals*.

- More broadly, Yb^{3+} upconversion occurs in many materials without the need for dimers. For example, upconversion could occur through the triplet annihilation mechanism (described above), energy transfer upconversion, or a cooperative mechanism as proposed by the authors [Rabouw et al. ACS Nano 12 (2018)]. While an appealing hypothesis, I find no evidence to suggest that dimers are involved.

Response: We thank the reviewer for the inspiring comments. We agree with that Yb^{3+} upconversion occurs in many pathways as you mentioned. However, we believe the other upconversion pathways can be excluded in this paper except cooperative sensitizing mechanism. Firstly, there lacks a proper singlet state for the triplet-triplet annihilation (TTA) as we have explained, excluding the probability of TTA. Meanwhile, it is reported that energy transfer upconversion (ETU) is an efficient and common pathway in fluoride upconverting nanoparticles. However, ETU process rely on an energy-matched intermediate level of the acceptor (in this paper, CsPbBr_3) as reported (*Chem. Soc. Rev.*, 2015, 44, 1608—1634, *Chem. Rev.* 2015, 115, 395-465). There is no such energy level in CsPbBr_3 detected nor reported as we know. In conclusion, we think the cooperative sensitizing is the most reasonable mechanism. On the basis of this hypothesis, the dimers seem to be of importance in the process as we explained in the last question.

- Two important aspects of the data I don't understand are how the authors calculate the 98.4% TET efficiency and why there is such a long rise time in the Figure 4 TA spectra, despite the short IRF (Fig. S15). The first is particularly important and not clear to me. I suggest the authors address these points to aid non-expert readers.

Response: We are sorry for the possible misunderstanding in the manuscript. Here, the dynamics of the dye is utilized to evaluate energy transfer time constant and efficiency. More specifically, for the sample of undoped NC-dye system, the excited state absorption (ESA) signal shows the intrinsic

dynamics of photo-excited IR783, which consists of S1-to-S0 transition (singlet decay) and T1-to-S0 transition (triplet decay). In comparison, the presence of Yb³⁺ in the NC core produces an additive decay channel of triplet energy transfer (TET) from surface-attached IR783 to Yb³⁺ doped in the NC core. To model such accelerated decay due to TET, the following equations are used:

$$k_{undoped} = k_S + k_T \quad (0.1)$$

$$k_{doped} = k_S + k_T + k_{TET} \quad (0.2)$$

where $k_{undoped}$, and k_{doped} refer to the decay rate of undoped NC-dye and doped NC-dye system. k_S , k_T and k_{TET} are denoted as the decay rate of singlet, triplet and TET rate, respectively. As a result, the TET rate can be extracted by:

$$k_{TET} = k_{doped} - k_{undoped} = \frac{1}{\tau_{doped}} - \frac{1}{\tau_{undoped}} \quad (0.3)$$

The efficiency of TET can be calculated by:

$$\eta_{TET} = \frac{k_{TET}}{k_S + k_T + k_{TET}} = \frac{k_{TET}}{k_{doped}} \quad (0.4)$$

The content of this response above is added to SI in order to eliminate possible misunderstanding.

Action: The formulas and the calculation process have been added to the revised supplementary information.

The authors should also address the following minor issues:

- A common challenge in lanthanide-based upconversion literature is rare earth contamination. Indeed, the 976 nm PL spectrum in Fig 3c looks like it may include some Er³⁺ or Tb³⁺ at 550 nm. The authors should comment about potential contamination.

Response: We agree with the comment. We have now pointed out the potential contamination in the revised manuscript.

Action: Added in the main text (Line 181) “It is noteworthy that the discrepancy of the spectra in Fig. 3c at ~550 nm may be caused by the potential Er³⁺ or Tb³⁺ contamination.”

- The CsPbBr₃ PL is inconsistent throughout the manuscript. For example, the PL in Figure 3c and Figure 3E are remarkably different. The N₂:O₂=1:0 data are intensely structured. Given the potential influence of rare earth contamination, this should be discussed.

Response: We thank the reviewer for the suggestion. The PL in Figure 3c was excited by 976 nm laser while the Figure 3e was excited by 804 nm laser. Indeed, the Er^{3+} or Tb^{3+} contamination may exist as above mentioned (and this discussion has been added in the revised manuscript as mentioned). In terms of the intensely structured PL, we repeated the measurement of the $\text{N}_2:\text{O}_2=1:0$ sample to eliminate the discrepancy. As a result, we do think it was caused by contamination and thank for your reminder. We have measured the same PL for several times and believe it was caused by random.

Action: We replaced the PL in Figure 3e by a newly measured result to eliminate the possibility of contamination.

- Additional photophysics characterization of the Yb^{3+} -doped and undoped CsPbBr_3 would improve the paper. Given evidence for lanthanide-based defect states in the CsPbBr_3 band gap [Milstein et al. Chem Mater 34 (2022)], you would expect to see a decrease in PL lifetime upon doping. I wonder how these defects, if present, may influence the upconversion process.

Response: We thank the reviewer for the comments. As you have expected, we also considered the impacts of the host nanocrystals after Yb^{3+} doping. Because of the weak PL of Yb^{3+} in the infrared region, we were unable to measure the lifetime. As a reference, our result of the ground-state bleach from transient absorption may help to understand the same process (Figure R5.3). As the results indicate, the ground-state bleach recovery is accelerated after Yb^{3+} doping as we expected. However, the phenomenon can be ascribed to the formation of the reported band-edge defects as well as the nonradiative energy transfer from exciton to Yb^{3+} . Meanwhile, we have not detected such a band-edge emission from defects for now. As a result, we think the influence of potential defect states is negligible in the upconversion process.

Figure R5.4. The transient absorption spectra of undoped and 40% Yb^{3+} doped CsPbBr_3 , excited under 343 nm, detected at 517 nm.

- The singlet lifetime analysis appears to be inconsistent. Figure S16 shows that the singlet is excited over ~ 4 ns (after ps excitation) and then decays from ~ 4 ns; while Figure S15 instead shows that the singlet is excited (after fs excitation) within 1 ps and decays by 1 ns. There may be a time-zero offset problem? Or the ps laser and detector have a long time constant? Plotting the instrument response function on the Figure S16 could resolve this. In addition, it isn't clear to me why the S15 transient absorption data lack a rise time, while the Figure 3a and b, Figure S14, and Figure S13 data do not have a rise time. Is this a feature of intersystem crossing?

Response: Thanks for your comment. We now further checked the data presented in Fig S13-S15. Data presented in Fig S13 and S14 were obtained by a nanosecond transient absorption (ns-TA) spectrometer (EOS, Ultrafast Systems). For this spectrometer, the pump laser is a femtosecond laser, while the probe laser is an ultra-continuous white laser with a pulse duration of about 1 ns. As a result, the instrumental response function (IRF) of this ns-TA spectrometer is larger than 1 ns, thus leading to a significant rising edge of the signal (~ 1 ns) in Fig S13 and 14. However, data from Fig S15 were obtained by femtosecond transient absorption (fs-TA) spectrometer with a temporal resolution of ~ 200 fs, so the rising edge is unobvious in Fig S15. As for the data in Fig S16, they were obtained as time-resolved PL (or PL dynamics) rather than TA dynamics and its IRF was given in Figure R5.5. The fitted time constants for both PL dynamics and TA GSB dynamics correspond well. As for the

time-zero offset problem, although the dynamics signal peaks at the timepoint of ~ 3 ns, it does not have any influence on fitting the decay dynamics. So, we present the original data without manual calibration of the time-zero point.

Figure R5.5. The instrument response function of the 785 nm pulsed laser.

Action: We have added the IRF of the 785 nm pulsed laser along with the lifetime in the revised Supplementary Fig S20.

- The authors claim the dye the ITO surface exhibits a 5-7 nm shift in optical absorbance due to solvent effects. Other authors have ascribed similar shifts to surface adsorption [Zou et al Nat Photon. 6 (2012)]. This may be additional evidence dyes are adsorbed onto the particle surface.

Response: We thank the reviewer for the comment. It is believed that the solvent effects of organic molecules commonly exist in solution, which can arise from several factors including the solvent polarity, viscosity and so on (*Joseph R. Lakowicz, Principles of Fluorescence Spectroscopy (third edition), chapter 6*). In our opinion, the surface adsorption to PeNCs can be seen as one type of solvent effects. Besides, the infrared spectroscopy results have been added to the revised manuscript (referred to Reviewer1, comment 4, Figure R1.4).

Action: We have the sentence “*The result from FTIR spectroscopy further proved the attachment (Supplementary Fig. S5)*” in the main text and the result from FTIR measurement was given in supplementary information.

• The authors conduct a nice control experiment showing the importance of sulfonic acid groups on sensitizing dye molecules. However, it's unclear if these dyes adsorb on the nanocrystal surface without the sulfonic acid groups. Without confirmation that the dye is on the nanocrystal surface, it's impossible to assess this experiment. The authors should prove that this alternative dye is on the nanocrystal surface. Optical absorption spectra the CsPbBr₃-Yb³⁺-IR780 conjugate could confirm this, for example.

Response: We thank the reviewer for the suggestion. In the revised manuscript, the absorption spectrum of CsPbBr₃-Yb³⁺-IR780 has now been provided in the supporting information (Figure R5.5). The strong absorption ranging from 650 to 850 nm indicated the adsorption of IR780 on the surface. Considering the negligible upconverting emission in CsPbBr₃-Yb³⁺-IR780, the importance of sulfonic acid group is further proved.

Figure R5.6. Absorption spectrum of CsPbBr₃-Yb³⁺-IR780

Action: Line 125, we have added the sentence “The absorption spectrum of IR780-CsPbBr₃:Yb³⁺ (Supplementary Fig. S6) indicated that IR780 have been adsorbed to the nanocrystals.” And the absorption spectrum has been added to the supplementary information.

• Throughout the manuscript, there are a range of potential typos in figures and captions that make review and interpretation challenging. These must be fixed. Some particularly relevant ones include:
o Figure 2d – what are the y-axis units? Are these spectra comparable? What is the illumination wavelength and irradiance for CsPbBr₃-Yb³⁺? What sample is being studied – is this the 40% Yb³⁺?

Response: We thank the reviewer for these reminders. In the revised manuscript, we have fixed the writing mistakes as far as we can find them.

Action: The units of Figure 2d have been added. The spectra in Figure 2d contains different information and they are explained in the main text as well as in the explanatory text. The blue curves (solid and dashed curve) are utilized to compared the exciton emission and the upconverting emission in shape and wavelength while the solid curves (blue, red and orange) aim to compared upconversion intensity under different conditions. It is noteworthy that we are reminded that the orange solid curve was totally covered by the red one so in the revised manuscript, the red solid curve was changed to red dashed curve. As explained in the main text (Line 135), the illumination wavelength peaks at 515 nm under 804 nm laser (power density= 8.4 W/cm²). Lastly, the sample studied throughout the paper is the 40% doping except when the influence of doping concentration was investigated.

o Figure 3e – what are the y-axis units?

Response: We thank the reviewer for the reminder. In the revised manuscript, the unit was added.

Action: The units of Y-axis of Fig.3e a.u. have been added.

o Figure 4c – the y-axis units should be included. Leaving these out obscures the data.

Response: We thank the reviewer for the reminder. In the revised manuscript, the units were added.

Action: The label and units of Y-axis of Fig. 4c Normalized ΔA (mOD) have been added.

o Figure S3 – what nominal doping concentration is this data for?

Response: We thank the reviewer for the reminder. Nominal doping concentration is defined as the Yb/Pb feeding ratio. Since the concept caused some misunderstandings, we listed a table (referred to Reviewer2, comment 2, Table R2.1) in the revised manuscript.

Action: Line 138, we have added the text while the actual doping concentration is 0.8% after the nominal concentration in original manuscript and added the table to the revised manuscript (**supplementary Table S1**).

- The quantum yields listed in Table S1 compare two-photon upconversion with dyes. However, the dyes are uncited and not representative of much more common dyes, such as Rhodamines, with photoluminescence quantum yields of ~ 50%. The authors should include more representative dyes in this table as well as references.

Response: We appreciate the reviewer for the suggestion. More dyes as well as upconverting nanocrystals have been listed to the table (referred to Reviewer1, comment 2, Table R1.1) in terms of their quantum yields, brightness and power density. Also, the references have been cited (we thank the reminder).

Action: The table have been fulfilled in the revised supplementary information as well as the cited references (**supplementary Table S2**).

Reviewer #1 (Remarks to the Author):

The quality of the revised manuscript has been improved and I recommend accept it.

Response: We appreciate the approval of our work and your thoughtful suggestions to improve this manuscript during the revision process.

Reviewer #2 (Remarks to the Author):

The article, in its present form, is much improved with respect to its previous edition. All the questions I raised have been well addressed or explained. I recommend it for publication without further revision.

Response: We thank the approval of our work and your kindly suggestions to improve this manuscript during the revision process.

Reviewer #3 (Remarks to the Author):

The article in its present form is much improved with respect to its previous edition. All my concerns have been well addressed or explained. I recommend it for publication without further change.

Response: We thank the approval of our work and your kindly review to improve this manuscript during the revision process.

Reviewer #4 (Remarks to the Author):

I have gone over the updated files and response letter. I find that the authors have thoughtfully considered and addressed (whenever possible) the comments and questions posed by the reviewers.

My concerns were particularly focused on issues of stability. The authors have added new stability measurements that provide insight, and have updated the text on this front. I also appreciate their further attempts and control measurements for untangling the roll of Yb in the energy transfer and upconversion process.

I support publication of the updated manuscript in Nature Communications.

Response: We appreciate the approval of our work and your helpful suggestions to improve this manuscript during the revision process. Besides, your kindly comments bring the authors much comfort.

Reviewer #5 (Remarks to the Author):

This revised communication from Zhang et al. is significantly improved in revision. The upconversion demonstration is impressive, occurs at relatively low irradiances and is an exciting compliment to the current literature, wherein halide perovskites commonly serve as a photoabsorber and single-to-triplet sensitizer. However, the photophysics are still missing some critical pieces of data needed to for readers to understand the results. These are already collected. Provided that plotting these and adding relevant discussion does not change the results, I strongly recommend publication.

Response: We sincerely appreciate the reviewer's thoughtful and insightful comments, as well as the constructive suggestions. We are also thankful for the reviewer's recommendation of publication. In the revised manuscript, we have incorporated the full kinetic traces of both IR783-CsPbBr₃:Yb³⁺ and IR783-CsPbBr₃ systems. Following the reviewer's suggestions, we have also revised certain figures and textual descriptions to enhance clarity and eliminate potential ambiguities. Furthermore, we have included new data demonstrating the existence of cooperative upconversion luminescence of CsPbCl₃:Yb³⁺ under 980 nm excitation at high laser irradiances (2825.3 W/cm²), which we believe provides compelling evidence for the existence of Yb³⁺-dimers in Yb³⁺-doped perovskite nanocrystals. We believe these revisions have substantially enhanced the clarity of the photophysical discussions and strengthened the overall conclusions. We therefore hope the revised manuscript will now be considered suitable for acceptance.

Photophysics control data:

In the authors' model, they propose that the IR-783 dye absorbs via the singlet state, undergoes intersystem crossing to form a triplet, which then transfer to Yb³⁺ in the CsPbX₃ nanocrystals. Here two excited Yb³⁺ atoms upconvert the energy forming an exciton. The photophysics in Figure 4A and S15 show that, without Yb³⁺ dopants, the IR-783 dye is excited and undergoes intersystem crossing to an excited triplet state. The singlet state is depopulated over ~ 3 ns. The photophysics in Figure 4B and S16 show that, with Yb³⁺ atoms, the IR-783 dye is excited, and the singlet state is depopulated over ~ 3 ns. The longer time region, during which the triplet state is presumably absent, is not shown. While the triplet state appears to be absent in the surface plot of Figure 4B, the optical densities are different. Indeed, there is a discrepancy in optical density between Figure 4 and SI Figures S15 and

S16, where the OD in the manuscript is half the OD in the SI, highlighting this importance.

To their credit, the authors added much of this requested data and noted that it was challenging to measure the triplet state in Yb-doped samples. I sympathize with this. The authors should still add this control data as detailed below for readers:

- The authors should add the TA spectra at these longer times (~100-180 ns) for IR783-Yb³⁺ CsPbBr₃ samples, even if it shows nothing. In other words, Figure S15B should be reproduced for the IR783-Yb³⁺ CsPbBr₃ samples. This should be included to allow reviewers to assess the authors' claims. To their credit, the authors do note that the triplet dynamics were challenging to observe in the Yb-doped sample.

Response: We thank the reviewer for the helpful comments. We agree with the suggestion to include the TA spectra of IR783- CsPbBr₃:Yb³⁺ samples at longer timescales. This data is now presented as **Figure R1.1** and included as **Figure S16** in the revised Supplementary Information. No distinct triplet signal was observed in the 50–100 ns range; if present, it was on the level of background noise.

Figure R1.1. Transient absorption spectra of IR783-CsPbBr₃:Yb³⁺ at (a) short and (b) longer timescale. Note that panel (b) uses milli-optical density (mOD), in contrast to optical density (OD) in panel (a), to provide an enlarged view of the measured signal.

Action: We have included **Figure R1.1** as **Figure S16** in the revised Supplementary Information.

- In addition, the authors claim that the triplet state lifetime in IR783-Yb-CsPbBr₃ is 1 ns, but the TA spectra in the SI (Figure S15) appear to show only singlet character. Because, as the authors have added, the triplet state has a much lower optical density, it is likely that this is challenging or

impossible to observe. The spectra in Figure S16 (and kinetic trace in Figure 4C) only show the decay of the singlet state, which the authors claim to be the decay of the triplet. This decay rate is the same as in Yb-CsPbBr₃. One explanation could be that the triplet state transfers at a rate much faster than the IRF. Are the authors measuring the singlet or triplet decay in this analysis? The authors should explain.

- This question is significant because this parameter is used to calculate the TET efficiency and rate. Would an IRF-limited triplet decay process change the triplet energy transfer efficiency calculation? The transfer rate may be much faster than reported.

- Likewise, Figure 4C still compares two entirely different features: the singlet decay of IR783-Yb-CsPbBr₃ (maybe this is also the triplet decay per the previous comment) vs. the triplet decay of IR783-CsPbBr₃. The IR783-Yb-CsPbBr₃ may contain no triplet signal at long times because it transfers. But this isn't shown in the data. In fact, this is obscured by normalizing the data in Figure 4C to different times.

The authors should include the full kinetic traces at 520 nm for both IR783-Yb-CsPbBr₃ and IR783-CsPbBr₃ over the same time range so that readers can compare in the SI. The authors added all data but these in this revision.

Response: We sincerely appreciate the insightful comment. In response, we have included the full kinetic traces at 520 nm for both IR783-Yb-CsPbBr₃ and IR783-CsPbBr₃ samples over the entire measured time range (**Figure R1.2**, or **revised Figure 4**). For IR783-CsPbBr₃ sample, two distinct kinetic components were observed—one in the short timescale (singlet) and another in the longer timescale (triplet). In contrast, only a short-timescale kinetic trace was observed for IR783-Yb-CsPbBr₃ sample.

We agree that the observed signal in IR783-Yb-CsPbBr₃ resembles the singlet TA signal of IR783-CsPbBr₃. However, its lifetime (~1.06 ns) is slightly longer than the singlet lifetime (0.75 ns) in IR783-CsPbBr₃. We attribute this to a possible overlap of singlet and triplet contributions with close lifetimes. Therefore, we use the observed 1.06 ns lifetime to estimate the triplet energy transfer (TET) kinetics and efficiency, yielding a TET efficiency of 98.4%.

Another possibility for the absence of a detectable long-timescale TA signal at longer time scale in IR783-Yb-CsPbBr₃ is that the triplet signal was quenched by the TET process to the level of background noise. Specifically, the triplet signal intensity at 92 ns decreased from 0.11 mOD in IR783-

CsPbBr₃ to 0.009 mOD in IR783-Yb-CsPbBr₃—within the noise threshold. Given that triplet optical density (OD) is proportional to the number of triplets, we estimate the TET efficiency using the relation $\eta = 1 - N_{DA}/N_D$, where N_D and N_{DA} represent the triplet populations in the absence and presence of the acceptor, respectively. This yields a lower-bound TET efficiency of 91.8%, consistent with the 98.4% estimate from the decay kinetics.

Taken together, these results suggest that the observed short-timescale signal in IR783-Yb-CsPbBr₃ is likely a combination of singlet and triplet contributions with similar lifetimes, and that the TET process proceeds with near-unity efficiency.

Figure R1.2. Triplet energy transfer dynamics from IR783 to CsPbBr₃:Yb³⁺ perovskite nanocrystals. **a**, Transient absorption spectrum of IR783-CsPbBr₃ nanocrystals. **b**, Transient absorption spectrum of IR783-CsPbBr₃:Yb³⁺ nanocrystals. **c**, TA decay of IR783-CsPbBr₃ nanocrystals and IR783-CsPbBr₃:Yb³⁺ nanocrystals over the entire measured timescale. **d**, Triplet decay (top) and singlet decay (bottom) of IR783-CsPbBr₃ nanocrystals compared with the decay of IR783-CsPbBr₃:Yb³⁺ nanocrystals; the solid curves represent the corresponding single-exponential

fitting results of the decay processes, detected at 522 nm, power density = 1.58 $\mu\text{J}/\text{cm}^2$.

Action: We have substituted the Figure 4 with Figure R1.2 in the revised manuscript. Besides, we have added the text “however, the long-lived component is absent (Supplementary Fig. S16)” in **Line 223**. We also modified the analysis in **Line 236**, “In the presence of Yb^{3+} , the estimated lifetime is about ~ 1.06 ns, slightly longer than the singlet lifetime of 0.75 ns in IR783-CsPbBr₃, which indicates the observed exclusive TA signal is probably a mixture of both singlet and triplet signal with close lifetimes. This measured lifetime was used to estimate the triplet lifetime.”

- Why are the transient absorption optical densities different in the SI vs. the manuscript?

Response: We sincerely appreciate the reviewer’s thoughtful and constructive comments, which helped us identify certain ambiguities in our presentation. Upon re-examination, we found that Figures 4A and S15, as well as Figures 4B and S16, were plotted using the same datasets.

Our analysis revealed that the singlet-state signals (<10 ns) in Figure 4A were significantly more intense than the triplet-state signals. To highlight the weaker triplet features, we had initially chosen an inappropriate optical density (OD) range of -0.005 to 0.08 , which caused all stronger signals (>0.08) to appear as saturated dark red. To correct this, we have carefully replotted Figure 4A in the revised manuscript using a more suitable OD range of -0.15 to 0.2 . This adjustment provides a clearer and more balanced representation of both the singlet and triplet-state signals (see **Figure R1.2**)

Action: We have substituted the Figure 4 with Figure R1.2 in the revised manuscript.

- Why is the TA signal of the Yb-containing sample lower than the samples without Yb?

Response: We sincerely appreciate the insightful comment. The primary factor is likely the variation in sample concentrations. In transient absorption measurements, excessively high absorbance is unfavorable; thus, although the two samples initially had identical concentrations, they were diluted to optimize the signal-to-noise ratio. This adjustment may have led to differences in concentration between samples.

Role of Yb-dimers

- Finally, the authors propose that Yb dimers are the species responsible for upconversion. In their rebuttal, the authors attempted upconversion in IR783-Yb-CsPbCl₃, which did not work. They claim that this demonstrates dimers must be active. However, the energy of two Yb³⁺ excitations (~ 2 eV) is less than the CsPbCl₃ band gap (~ 3 eV). While I appreciate the additional experiments, this upconversion process is not active in CsPbCl₃ because upconversion would require three Yb atoms. Thus, this experiment does not prove Yb-dimers are or aren't active.

It may indeed be the case that Yb dimers are important, but the authors do not show that dimers are necessary or involved. However, the authors do demonstrate that surface Yb is important for dye binding. These surface Yb species cannot form dimers. This discrepancy suggests that dimers may not be active.

This comment isn't to say that Yb dimers are or aren't involved. They may be. As written, the manuscript claims that dimers are involved without evidence. This could be addressed by adding new data showing that Yb dimer species are involved or by adjusting the manuscript to emphasize that upconversion could be enhanced by Yb dimers.

Response: We thank the reviewer for the detailed comments. We believe Yb³⁺ dimers are involved in the upconversion process, based on the following observations: (i) The presence of Yb³⁺ ions is essential for upconversion luminescence in IR783-CsPbBr₃: Yb³⁺ nanocrystals. In the absence of Yb³⁺, no exciton upconversion luminescence is observed in IR783-CsPbBr₃ under 808 nm excitation. (ii) The upconversion luminescence intensity is strongly dependent on the concentration of Yb³⁺ ions in IR783-CsPbBr₃: Yb³⁺ nanocrystals. (iii) Direct excitation of Yb³⁺ at 980 nm (~1.3 eV) can induce exciton upconversion luminescence in CsPbBr₃: Yb³⁺ nanocrystals, which have a bandgap of ~2.4 eV. According to the law of energy conservation, two excited Yb³⁺ ions are required to provide sufficient energy to generate one exciton. Taken together, these results suggest that two Yb³⁺ ions must simultaneously participate to enable exciton upconversion luminescence. The most probable scenario is the involvement of Yb³⁺ dimer—a pair of Yb³⁺ ions in close proximity within the lattice.

To further address the reviewer's concern, we performed high-power excitation of CsPbCl₃:Yb³⁺ nanocrystals at 980 nm (power density=2825.3 W/cm²), and observed upconversion luminescence peaked at about 500 nm (~ 2.5 eV) (**Figure R1.3**). The photon energy of luminescence is clearly below the bandgap (~ 3 eV), ruling out the exciton upconversion luminescence. We also calculated

the self-convolution of the Yb^{3+} luminescence spectrum in $\text{CsPbCl}_3:\text{Yb}^{3+}$ nanocrystals (shown as the red solid line), using the following formula:

$$I_{\text{self-conv}}(E) = \int_{-\infty}^{+\infty} I(E') I(E - E') dE'$$

where I denote the luminescence intensity, E and E' denote the corresponding energy of different wavelengths from Yb^{3+} emission. As a result, the observed upconversion luminescence generally agrees well with the self-convolution of Yb^{3+} emission, which can be interpreted as cooperative luminescence from Yb^{3+} dimers. The broadband luminescence background may arise from the involvement of defect-related energy levels. This experimental observation gives compelling evidence for the existence of Yb^{3+} dimers in Yb^{3+} doped perovskite nanocrystals.

Figure R1.3. Upconversion luminescence from $\text{CsPbCl}_3:\text{Yb}^{3+}$ nanocrystals under high laser power density ($\lambda_{\text{EX}}=980$ nm, power density= 2825.3 W/cm²), in comparison with the self-convolution of Yb^{3+} luminescence.

Action: We have added the Figure R1.3 as Figure S20 in the revised Supplementary Information, and added text “*The observation of cooperative upconversion luminescence from Yb^{3+} -dimers in perovskite nanocrystals ($\text{CsPbCl}_3:\text{Yb}^{3+}$) gives evidence for the existence of such defects (Supplementary Fig. S20)*” to the main text (Line 252).

As a minor comment, Figure S5 caption has a typo and should be $\text{CsPbBr}_3:\text{Yb}^{3+}$ for the top panel.

Response: We appreciate the kind comment; we have corrected the typos in the revised Supplementary Information.

Action: We have corrected the mistake in Fig.S5.